# Antibody-guided in vivo imaging of *Aspergillus fumigatus* lung infections during antifungal azole treatment

Sophie Henneberg[1], Anja Hasenberg [1], Andreas Maurer [2], Franziska Neumann[1], Lea Bornemann [1], Irene Gonzalez-Menendez[3], Andreas Kraus[1], Mike Hasenberg[1], Christopher R. Thornton [4], Bernd J. Pichler[2], Matthias Gunzer [1,5 ✉] & Nicolas Beziere [2 ✉]

Invasive pulmonary aspergillosis (IPA) is a life-threatening lung disease of immunocompromised humans, caused by the opportunistic fungal pathogen *Aspergillus fumigatus*. Inadequacies in current diagnostic procedures mean that early diagnosis of the disease, critical to patient survival, remains a major clinical challenge, and is leading to the empiric use of antifungal drugs and emergence of azole resistance. A non-invasive procedure that allows both unambiguous detection of IPA and its response to azole treatment is therefore needed. Here, we show that a humanised *Aspergillus*-specific monoclonal antibody, dual labelled with a radionuclide and fluorophore, can be used in immunoPET/MRI in vivo in a neutropenic mouse model and 3D light sheet fluorescence microscopy ex vivo in the infected mouse lungs to quantify early *A. fumigatus* lung infections and to monitor the efficacy of azole therapy. Our antibody-guided approach reveals that early drug intervention is critical to prevent complete invasion of the lungs by the fungus, and demonstrates the power of molecular imaging as a non-invasive procedure for tracking IPA in vivo.

[1] Institute for Experimental Immunology and Imaging, University Hospital, University of Duisburg-Essen, Essen, Germany. [2] Werner Siemens Imaging Center, Department of Preclinical Imaging and Radiopharmacy, Eberhard Karls University, Tübingen, Germany. [3] Department of Pathology and Neuropathology, Eberhard Karls University, Tübingen, Germany. [4] ISCA Diagnostics Ltd. and Biosciences, College of Life and Environmental Sciences, University of Exeter, Exeter, UK. [5] Leibniz-Institut für Analytische Wissenschaften ISAS -e.V, Dortmund, Germany. ✉email: matthias.gunzer@uni-due.de; nicolas.beziere@med.uni-tuebingen.de

*A*spergillus fumigatus is an abundant environmental mold, with air-borne spores that are frequently inhaled. The fungus is an opportunistic pathogen, and while the immune system of healthy individuals is highly effective at eliminating infective spores from the lung, patients with impaired immunity, especially those with hematological malignancies, prolonged neutropenia, or recipients of hematopoietic stem cell or solid organ transplants, are at elevated risk of developing invasive pulmonary aspergillosis (IPA), a rapidly progressive and often fatal lung disease caused by germinated spores of the pathogen. The disease is also increasingly reported in patients with underlying respiratory diseases such as severe asthma and COPD, and as co-infections in patients with severe influenza and with Covid-19 coronavirus[1–3], regardless of their immune system status[4,5]. The high mortality rate of IPA[6] is exacerbated by the lack of rapid, specific and sensitive diagnostic tests. Since the symptoms of the disease (fever and chills, haemoptysis, shortness of breath, chest pain, headaches) are nonspecific, diagnosis relies on culture of the pathogen from invasive biopsy[7], or detection of biomarkers in serum or in bronchoalveolar lavage fluid (BALf) recovered during invasive bronchoscopy[8]. Attempts have been made to improve noninvasive diagnosis using computed tomography (CT)[8] or magnetic resonance imaging (MRI)[9] of the chest, but radiological indicators of infection are not pathognomonic for IPA. Despite this, a radiological abnormality in a chest-CT is used in many centers as a trigger for initiating antifungal drug treatment in a febrile patient unresponsive to antibiotics[10]. Inappropriate or delayed treatment of IPA, driven by imprecise diagnostic procedures, negatively impacts on patient morbidity and mortality, and is contributing to the emergence of azole resistance in clinical strains of *A. fumigatus*[11,12]. Therefore, improvements in IPA diagnosis are needed that allow non-invasive detection of lung infection in vivo, and enable monitoring of disease responsiveness to antifungal drug treatments.

The use of targeted contrast agents in molecular imaging procedures such as positron emission tomography (PET) or single photon emission computed tomography (SPECT) holds enormous promise in addressing this unmet need[13,14]. While PET has proven its imaging capabilities, leading to its widespread use in oncology, only very few studies have tried to capitalize on its potentially high sensitivity and selectivity for infectious disease diagnosis beyond [18F]FDG. For invasive mycoses, radiolabeled antifungal drugs[15] or fungal siderophores[16,17] have been attempted, with mixed preclinical success, but their translation to the clinic appears problematic due to their lack of specificity. This limitation is overcome by the use of radiolabeled monoclonal antibodies, whose specificity allows targeted detection of individual pathogens when applied in immunoPET. Recently, we have focused on an *Aspergillus*-specific mAb, JF5[18], which binds to an antigen expressed on the surface of actively growing hyphae of *A. fumigatus*. While initial investigations using [64]Cu-radiolabeled murine JF5 showed promising results for in vivo imaging of IPA in a mouse model of the disease[19], further improvements in JF5-guided PET/MR imaging were met through its humanization[20], enabling translation of the technology to the clinical setting.

An important but unexplored aspect of molecular imaging is its use as a treatment monitoring tool for invasive fungal diseases. While attempts have been made to use circulating biomarkers to monitor responsiveness of IPA to antifungal treatment[21], it is well-documented that the sensitivities of diagnostic biomarker tests for the disease, such as the *Aspergillus* galactomannan (GM) ELISA and fungal β-D-glucan assay, are markedly influenced by mold-active azoles such as voriconazole (VCZ), both in model systems and in humans[22,23]. Furthermore, there is no data currently available on the relationship between fungal biomass in the lung and amounts of circulating biomarkers in serum or BALf,

and so it is not possible to determine unequivocally whether lung infection has been resolved without a visual appraisal using nonspecific chest-CT.

A key step to using immunoPET/MRI as a treatment monitoring tool is to establish whether it is sufficiently accurate to allow changes in *Aspergillus* load in the lung to be quantified in response to antifungal drug treatment. To this end, we set out here to (1) establish whether accumulation in the lung of the humanized JF5 (hJF5) radiotracer during *A. fumigatus* infection of neutropenic mice is directly related to fungal load, and (2) whether changes in the fungal load in the lung coincident with VCZ treatment can be monitored quantitatively in vivo. To achieve these goals, we developed a dual-labeled hJF5 variant bearing both a radionuclide and a fluorophore, [[64]Cu]Cu-NODAGA-hJF5-DyLight650 (hereafter [64]Cu-hJF5-DyLight650), which allowed us to co-localize and to co-quantify the pathogen and tracer in the lung both in vivo using broad resolution immunoPET/MRI, and ex vivo using high-resolution 3D light sheet fluorescence microscopy. Using this approach, we show that immunoPET/MRI can be used to monitor disease progression during treatment with an azole drug, and that early intervention with the drug is critical to preventing *A. fumigatus* infection.

## Results

**Characterization of the dual-labeled [64]Cu-hJF5-DyLight650 antibody**. The elution profile of the double-conjugated and radiolabeled antibody showed a single peak corresponding to the molecular weight of the antibody, with no detectable aggregates and only a small (<5%) dimer peak (Supplementary Fig. 1). To exclude detrimental effects of the chelator and fluorophore conjugation to antigen binding, the reactivity of the double-conjugated antibody was determined by ELISA using serial dilutions of the respective antibodies (Supplementary Fig. 2). We found only slightly reduced antigen binding of the antibody after single (NODAGA) and double (NODAGA and DyLight) conjugations, compared to the unconjugated antibody. Fitting of the binding curves revealed apparent Kd values of 67 pM (unmodified antibody), 59 pM (NODAGA-conjugated), and 112 pM (double-conjugated).

**Light sheet fluorescence microscopy of *A. fumigatus* in healthy and neutropenic mice**. To visualize the proliferation of *A. fumigatus*tdTomato in healthy and neutropenic animals, their lungs were removed 30 min, 6, 24, and 48 h postinfection. Following fixation and clearing, the lungs were imaged using Light Sheet Fluorescence Microscopy with the fungus being detectable via the endogenous tdTomato fluorescence signal (Fig. 1a). This revealed that in both immunosuppressed (neutropenic) and immuno-competent animals, there was patchy distribution of the pathogen 30 min after administration of the spore inoculum (Fig. 1b). Infestation of the lungs of neutropenic mice had occurred by 6 h p.i., with further proliferation of the pathogen thereafter until termination of the experiment. In comparison, healthy animals receiving the same spore inoculum showed a markedly reduced proliferation of the pathogen during the first 6 h postinfection, and a complete cessation by 48 h (Fig. 1b). High-resolution multiphoton microscopy of the same samples showed that the neutrophil depletion permitted substantial hyphal outgrowth by 24 h which was further increased by 48 h. In contrast, the presence of neutrophils completely blocked hyphal development, and led to total recovery of previously infected lung tissues, such that by 48 h it was morphologically indistinguishable from the non-infected lung (Fig. 1c).

Quantification showed that while the initial loads of *A. fumigatus* were comparable at the earliest time point (30 min p.i.), differences

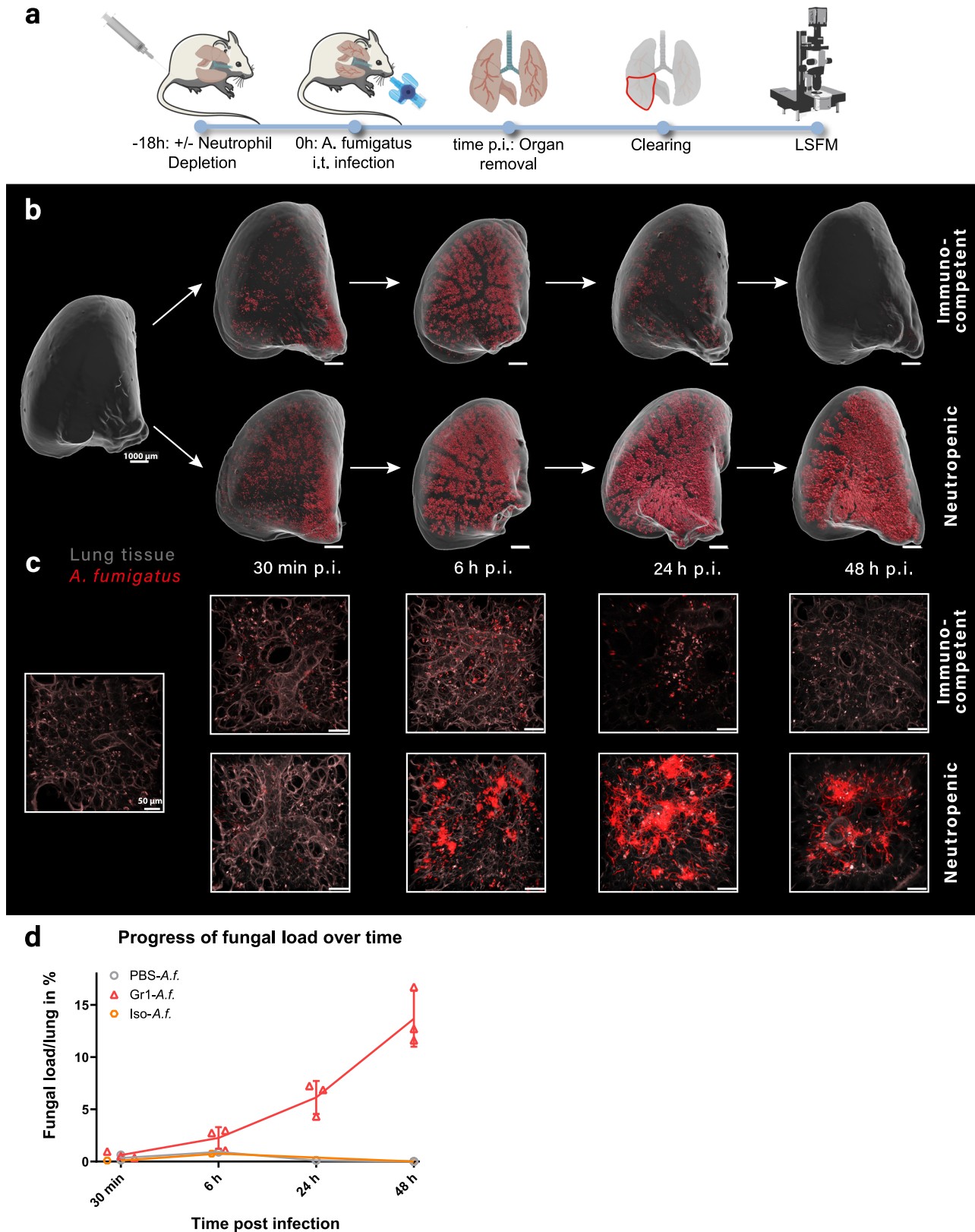

in loads between healthy and neutropenic mice were evident as early as 6 h p.i., and continued to rise over the period of observation (Fig. 1d). Hence, a functional neutrophil response controls fungal spread within the first hours of infection and effectively suppresses hyphal development.

**In vivo immunoPET/MR of *A. fumigatus* infection and treatment monitoring**. In previous studies, we showed the potential of the murine mAb JF5 and its humanized counterpart hJF5[19,20] to detect *A. fumigatus* lung infection in vivo using PET/MR. To determine whether this approach could be applied to treatment

**Fig. 1 Development of invasive pulmonary aspergillosis under healthy and immunosuppressed conditions. a** Schematic experimental workflow. Neutropenic groups were depleted with Gr-1 antibody 18 h prior to intratracheal (i.t.) inoculation with *Aspergillus fumigatus*tdTomato conidia. Lungs were subsequently removed at different time postinfection (p.i.), cleared and imaged using light sheet fluorescence microscopy (LSFM). **b** Lung tissue (gray) and fungal biomass (red) is depicted for time points 0, 0.5, 6, 24, and 48 h during disease progression. Images were constructed using surface tool from Imaris. Scale bar = 1 mm. **c** Magnified two-photon microscopy images of the same lungs as in **b**, with lung tissue (gray), and fungal biomass (red). Scale bar = 50 μm. **d** Plot of fungal load per whole lung lobe for the different treatments over time. Sham-depleted, *Aspergillus fumigatus* (A.f.) infected group (PBS-*A.f.*, gray circle, n = 2 per time point), Gr-1 neutrophil-depleted and infected group (Gr1-*A.f.*, red triangles, n = 3 per time point), isotype-depleted and infected group (Iso-*A.f.*, orange squares, n = 1 per time point). Results are plotted as individual values with average and standard deviation where applicable.

monitoring, we investigated the growth of *A. fumigatus*tdTomato in the lungs of healthy or neutrophil-depleted mice in response to treatment with VCZ, a triazole drug used in humans for primary therapy of IPA[24]. We first determined the VCZ susceptibility of our *A. fumigatus* strains (the wild-type strain ATCC 46645 and the genetically modified derivative *A. fumigatus*tdTomato). Minimum inhibitory concentration (MIC) determination for VCZ by visual reading, as suggested in the EUCAST guidelines[25], resulted in 0.5 μg/ml for ATCC 46645 as well as for *A. fumigatus*tdTomato strain (Supplementary Fig. 3 and Supplementary Table 1), similar to published data[25]. To corroborate these results with a quantitative reading method we analyzed the 96-well plates spectrophotometrically following the suggestion of Meletiadis et al.[26]. The results of the spectrophotometrical analysis yielded a VCZ MIC of 0.5 μg/ml for ATCC 46645 and of 0.25 μg/ml for strain *A. fumigatus*tdTomato (Supplementary Fig. 4 and Supplementary Table 2). These confirmed our previous findings using visual appraisals, and are consistent with the MIC range published in the works of Lass-Flörl and Meletiadis et al. In in vivo experiments, we first tested i.v. administration of VCZ 24 h postinfection and tracer injection (Fig. 2a), the time period corresponding to widespread colonization of the neutropenic lung (Fig. 1b). Images acquired 24 and 48 h after $^{64}$Cu-hJF5 and inoculum injection, showed no differences in the qualitative distribution of the radiotracer in the lungs of animals with VCZ treatment and those without (Fig. 2b). Indeed, significant accumulation was seen in the lungs of both groups of animals due to specific binding of the antibody tracer to its hyphal-bound antigen, and in the spleen due to soluble circulating antigen, a phenomenon reported previously[19,20]. As expected, a $^{64}$Cu signal could also been seen in the heart, indicating prolonged circulation of the radiotracer. Quantification of the in vivo $^{64}$Cu PET signal in the lungs, expressed as %ID/cc (Fig. 2c), confirmed the distributions seen qualitatively (Fig. 2b). No significant differences in tracer uptake in the lungs were found between VCZ-treated and non-treated animals at 24 h (10.0 ± 2.0%ID/cc and 9.9 ± 1.7%ID/cc, respectively) and 48 h (11.1 ± 1.7%ID/cc and 12.9 ± 2.7%ID/cc, respectively). At 48 h, a significant increase in lung uptake of $^{64}$Cu-hJF5 could be seen in both groups compared to the control animals (6.5 ± 1.7%ID/cc) and, while the lung PET signal remained stable through time in control animals, the signal in infected animals increased significantly over the 48 h period, reaching a contrast ratio of ~2. Ex vivo biodistribution analysis of the tracer in different organs at 48 h (Fig. 2d) after perfusion by 0.4% PFA showed no significant differences in lung uptake between VCZ-treated and non-treated animals (38.4 ± 8.9%ID/g and 41 ± 18% ID/g, respectively), but with both significantly greater compared to control (uninfected) animals (3.9 ± 1.2%ID/g). Although no differences could be seen in heart or muscle uptake between groups, differences in the accumulation of $^{64}$Cu-hJF5 could be seen in the spleens of control and infected animals, with infected animals displaying an approximate twofold increase in uptake regardless of VCZ treatment. The circulating blood levels of the $^{64}$Cu-hJF5 tracer, obtained from blood sampling performed

directly before animal perfusion, were similar in the infected cohorts (31.9 ± 5.8%ID/cc and 36.4 ± 2.8%ID/cc in untreated and azole-treated groups, respectively), which was higher than in the control group (19.7 ± 2.6%ID/cc) indicating the long half-life of the full-length IgG in the bloodstream, but also possibly due to binding of the radiolabeled antibody to circulating antigen. Collectively, these data show that treatment with VCZ following establishment of lung infection by *A. fumigatus*, has no measurable effect on progression of disease in the neutropenic lung within the timeframe of the experiment.

**Dual-label in vivo immunoPET and ex vivo LSFM.** The lack of effect of VCZ led us to investigate whether the macroscopic resolution of PET/MR was insufficient to discriminate subtle differences in fungal burden consequent with antifungal treatment. We therefore set out to investigate the correlation between tracer accumulation and fungal burden in the lung using a combined PET/MR and LSFM imaging approach. To this end, we developed a dual-labeled derivative of hJF5, with $^{64}$Cu as the radioactive PET reporter, and DyLight650 as a fluorescent reporter for optical imaging. After infecting neutropenic mice with *A. fumigatus*tdTomato and immediately injecting $^{64}$Cu-hJF5-DyLight650, we determined whether VCZ administered 24 h after infection had an effect on the spatial distribution of the antibody 48 h after infection, and its co-localization with the fungus in infection sites at a macroscopic and microscopic level (Fig. 3a). Imaging the same lungs with both in vivo immunoPET/MR and ex vivo LSFM (Fig. 3b, c) revealed, qualitatively, complete overlap between fungal growth (depicted by red tdTomato fluorescence) (Fig. 3c), hJF5 labeling of hyphal antigen via the green DyLight650 fluorophore (Fig. 3c), and macroscopic distribution in vivo via the radionuclide $^{64}$Cu in immunoPET/MR (Fig. 3b). Confocal microscopy of the $^{64}$Cu-hJF5-DyLight650 molecule and tdTomato fluorescence further showed co-localization of fungal foci and the antibody at the microscopic level (Fig. 3d).

**In vivo and ex vivo volumetric quantification of fungal invasion.** In order to quantify fungal load in the lungs in vivo, we investigated how thresholding of the PET signal at different radiotracer accumulation values might enable accurate measurements of the volume occupied by the pathogen to be made. The ratio of the infection volume, as measured in PET, to the overall lung volume measured in MRI is shown in Fig. 4a. While a low threshold of 5%ID/cc, under the average %ID/cc in healthy animals, provided a significant difference between healthy controls and diseased animals, it nevertheless gave a strong false negative result, with ~50% of lungs supposedly infected in control animals. The variability was also high, since the threshold value was close to the background level. In contrast, a threshold of 10%ID/cc, selected to be significantly above the average PET tracer accumulation of a healthy lung at 48 h (6.5 ± 1.7%ID/cc), showed a significant difference between control and infected animals (3.2 ± 4.5% compared to 61 ± 4.5%; $p < 0.005$). However, VCZ-treated animals did not show any difference when compared to non-

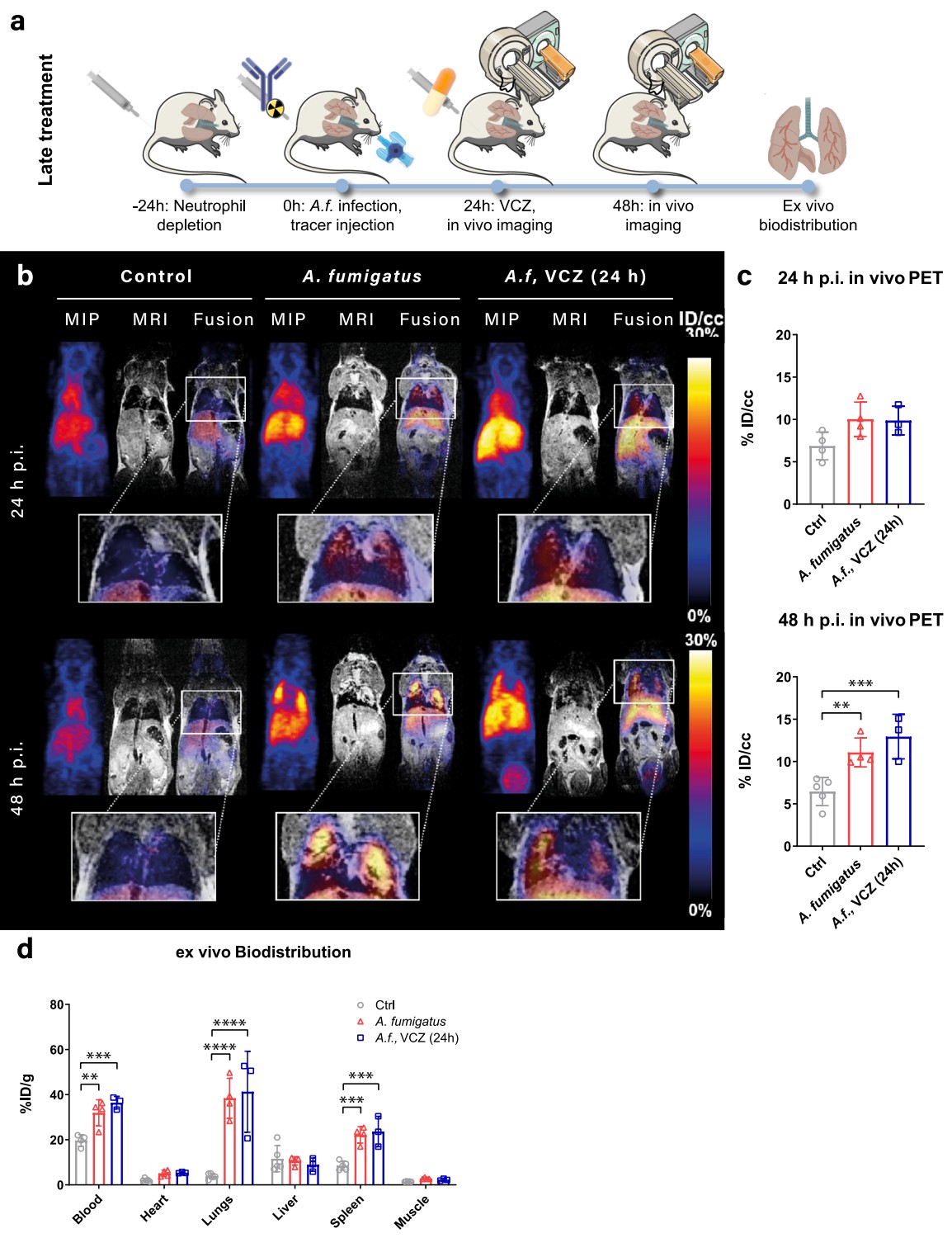

**a** Late treatment

-24h: Neutrophil depletion — 0h: *A.f.* infection, tracer injection — 24h: VCZ, in vivo imaging — 48h: in vivo imaging — Ex vivo biodistribution

**b**

Control | *A. fumigatus* | *A.f*, VCZ (24 h)
MIP  MRI  Fusion | MIP  MRI  Fusion | MIP  MRI  Fusion

24 h p.i.

48 h p.i.

ID/cc 30% / 0%

**c** 24 h p.i. in vivo PET

48 h p.i. in vivo PET

**d** ex vivo Biodistribution

Ctrl / *A. fumigatus* / *A.f.*, VCZ (24h)

Blood  Heart  Lungs  Liver  Spleen  Muscle

treated animals, with an infected volume of 54 ± 8%. A further increase in threshold volume to 15%ID/cc, selected to match the maximal PET signal obtained from healthy lung tissue in vivo (14.3 ± 1.6%ID/cc on average) provided an infected volume of 0% for control animals with no difference between non-treated and treated infected animals (22 ± 5% and 21 ± 10%, respectively). An even higher threshold of 20%ID/cc provided negligible infection volumes of <5% in infected animals but did not yield significant differences between groups. In addition, we performed a receiver operating characteristic analysis (Supplementary Fig. 5). This showed that 10 and 15%ID/cc were optimal thresholding values

allowing differentiation between healthy and infected animals with an area under curve (AUC) of 1, whereas the threshold values 5 and 20%ID/cc showed an AUC under 0.9. A similar thresholding approach was applied to LSFM images (Fig. 4b), comparing the volumes occupied by *A. fumigatus*tdTomato. While control animals did not present any fluorescent signal, similar values for volume occupancy by the pathogen and by the ⁶⁴Cu-hJF5-DyLight650 tracer were found in infected animals, and in VCZ-treated and non-treated infected animals, with ~20% of the lung displaying fluorescence signals (and hence active infection). Co-localization of *A. fumigatus* and hJF5, as measured in LSFM, yielded ~75%

**Fig. 2 ImmuoPET/MR imaging of invasive pulmonary aspergillosis in response to voriconazole (VCZ) treatment. a** Experimental workflow depicting neutrophil depletion, *Aspergillus fumigatus* (*A.f.*) infection, tracer injection, voriconazole administration (VCZ) and in vivo imaging. **b** ImmunoPET/MR images of PBS-deposited animals (Control, left column, $n = 4$), *A. fumigatus*-infected animals without VCZ treatment (*A. fumigatus*, middle column, $n = 4$), and with late-stage VCZ treatment (*A.f.* VCZ (24 h), right column, $n = 3$). Images acquired 24 h after inoculation (p.i.) are presented in the top row, the bottom row are images acquired 48 h p.i.. MIP is a Maximum Intensity Projection of the PET image; MRI is a T2-weighted MRI single slice MRI image; Fusion is a single slice PET image overlaid on the corresponding MRI slice. Magnification of the fusion image in the lung area is displayed below. **c** Quantification of $^{64}$Cu-hJF5 accumulation in the lung tissue at 24 h (top) and 48 h (bottom) after radiotracer injection in control (Ctrl), infected (*A. fumigatus*) and voriconazole treated animals (*A.f.*, VCZ (24 h)). 48 h p.i.: Ctrl vs. *A. fumigatus*, $p = 0.0152$; Ctrl vs. *A.f.*, VCZ (24 h), $p = 0.0033$. **d** Ex vivo Biodistribution data of $^{64}$Cu-hJF5 in major organs 48 h after radiotracer injection obtained after perfusion of the animal by 0.4% PFA. Blood: Ctrl vs. *A. fumigatus*, $p = 0.0032$; Ctrl vs. *A.f.*, VCZ (24 h), $p = 0.0002$. Lungs: Ctrl vs. *A. fumigatus*, $p < 0.0001$; Ctrl vs. *A.f.*, VCZ (24 h), $p < 0.0001$. Spleen: Ctrl vs. *A. fumigatus*, $p = 0.0005$; Ctrl vs. *A.f.*, VCZ (24 h), $p = 0.0004$. Results are plotted as individual values with average and standard deviation and are expressed as percent injected dose per cubic centimeter (%ID/cc) for PET and injected dose per gram (ID/g) for Biodistribution. $p$ values were generated using one way ANOVA (**c**) or two way ANOVA (**d**) with Tukey's multiple comparison test (*$p < 0.05$; **$p < 0.01$; ***$p < 0.001$; ****$p < 0.0001$).

correlation in both treated and non-treated groups (Fig. 4c), which was further supported by good correlation between the volume occupied by *A. fumigatus* in the diseased lung and hJF5-DyLight650 accumulation, regardless of VCZ treatment (Fig. 4d). This indicated that DyLight-hJF5 faithfully reports *A. fumigatus* invasion in the organ. When the in vivo immunoPET/MRI and ex vivo LSFM data were combined with a 15%ID/cc threshold for determination of disease volume (obtained by thresholding based on maximum signal intensity in healthy lungs), correlation between immunoPET/MR and LSFM volumetric findings per animal (Fig. 4e) showed a linear correlation ($r^2 = 0.8978$ and a slope of $0.87 \pm 0.17$), which was not the case with a threshold of 10% ($r^2 = 0.1832$). Hence, dual-label imaging with an hJF5-targeted strategy allows precise determination of the extent of fungal loading at a macroscopic and microscopic level.

**Early onset therapy is critical for infection control**. To treat IPA successfully, it is essential to discover lung infection as early as possible during disease progression, so as to allow timely treatment with mold-active drugs. To compare the effectiveness of early- and late-stage treatment on disease progression, we evaluated animals treated with VCZ at 3 and 24 h postinfection using immunoPET/MR. We had already established using LSFM (Fig. 1) that by 6 h postinfection the pathogen had begun to invade the entire lung, albeit at a low level, while widespread establishment of the pathogen had occurred by 24 h. We therefore chose the time points 3 and 24 h postinfection as the points at which to administer VCZ, so that we could then determine the effectiveness of early and late antifungal treatment start on disease progression using immunoPET/MR (Fig. 5a). At 24 h postinfection, immunoPET/MR images of disease progression under the two different VCZ treatment regimens (3 h + 24 h (early/late-stage) or 24 h only (late-stage)) showed minimal differences in pathogen establishment (Fig. 5b). This was confirmed by the quantitative in vivo PET data at 24 h postinfection, which showed no significant difference in tracer uptake for the control group (infected but untreated mice) ($9.8 \pm 1.6$%ID/cc), mice treated with VCZ at 24 h ($9.5 \pm 1.5$%ID/cc), and mice treated with VCZ at 3 and 24 h postinfection ($8.3 \pm 0.6$%ID/cc) (Fig. 5c). However, 48 h after infection, qualitative differences in $^{64}$Cu-hJF5 accumulation in infected animals under late- or early/late-stage treatment were pronounced, with the early/late-stage treated animals showing very limited uptake of tracer in the lungs, similar to that seen in non-infected animals (Figs. 3b and 5b). In contrast, there was extensive uptake of the tracer in animals receiving late-stage treatment, similar to that seen in infected, untreated animals (Figs. 3b and 5b). The quantitative in vivo PET data acquired at 48 h, showed that animals receiving late-stage treatment had slightly increased uptake of the tracer compared to levels at 24 h postinfection ($11.9 \pm 2.2$%ID/cc, Fig. 5c), while animals receiving the early/late-stage treatment displayed significantly ($p < 0.005$) reduced accumulation

($7.3 \pm 0.5$%ID/cc) compared to the untreated animals, and those receiving late-stage VCZ treatment. Using the volumetric approach described above, with a threshold of 10%ID/cc, significant differences ($p < 0.005$) were already apparent at 24 h postinfection between animals receiving late-stage vs. early/late-stage VCZ treatment ($40 \pm 14$% and $22 \pm 6$%, respectively, Fig. 5d) which was not the case when using a 15%ID/cc threshold. At 48 h postinfection, the differences in tracer uptake between the two groups was even more pronounced ($61 \pm 14$% for animals receiving late-stage VCZ, and $11 \pm 4$% for animals receiving early/late-stage VCZ treatment). Using a 10%ID/cc threshold, a significant difference between animals receiving early/late-stage treatment and the animals receiving late-stage or no treatment was found although with high volume occupation values ($11 \pm 4$%ID/cc vs. $61 \pm 14$ and $58 \pm 12$%ID/cc, respectively, $p < 0.0001$). When a 15%ID/cc threshold was applied to the PET data as in the experimental results presented in Fig. 4d, e, no positive (infected) volume was exhibited in animals receiving early/late-stage treatment, while the animals receiving late-stage treatment displayed a $25 \pm 17$% infected lung volume (Fig. 5c). Ex vivo biodistribution data obtained after perfusion of the animals (Supplementary Fig. 6) were in accordance with the in vivo findings. They showed a significantly reduced lung accumulation of $^{64}$Cu-hJF5 in animals receiving early/late-stage VCZ treatment compared to the late-stage animals ($12 \pm 6$%ID/g vs. $35 \pm 14$%ID/g, respectively, $p < 0.001$). Liver accumulation showed similar trends, with $6.9 \pm 2.6$%ID/g for early/late-stage VCZ-treated animals, and $20 \pm 13$%ID/g for late-stage treated animals ($p < 0.05$). While no other organs showed measurable differences in the accumulation and retention of $^{64}$Cu-hJF5 across the two treatment groups, animals receiving early/late-stage VCZ treatment displayed an elevated, albeit non-significant, amount of blood pool tracer. In addition, we have performed hematoxylin and eosin (H&E) stain as well as Grocott methenamine silver stain on lung samples to observe the tissue damage and extent of the fungal invasion (Supplementary Fig. 7). The lungs of animals infected with A. fumigatus but not treated showed massive central bronchial infiltration, multiple large granulomas containing hyphae and spores, cavitations, necrosis, and destruction of the respiratory epithelium. Lungs receiving late treatment showed severe damage, with multiple and large *Aspergillus* granulomas containing numerous and large hyphae. *Aspergillus* spores were detected throughout the respiratory parenchyma. We observed large necrotic areas around the bronchi and severe destruction of the respiratory epithelium. Angioinvasion was also detected, with necrosis of the surrounding tissue. The lungs of mice receiving early treatment showed multiple small bronchio-centric inflammatory infiltrates with no significant damage of the respiratory epithelium. Few and small *Aspergillus* granulomas were identified, which contained exclusively spores. No hyphae were detected. The lungs of the control group (PBS instilled) showed normal histology, without pathological alterations.

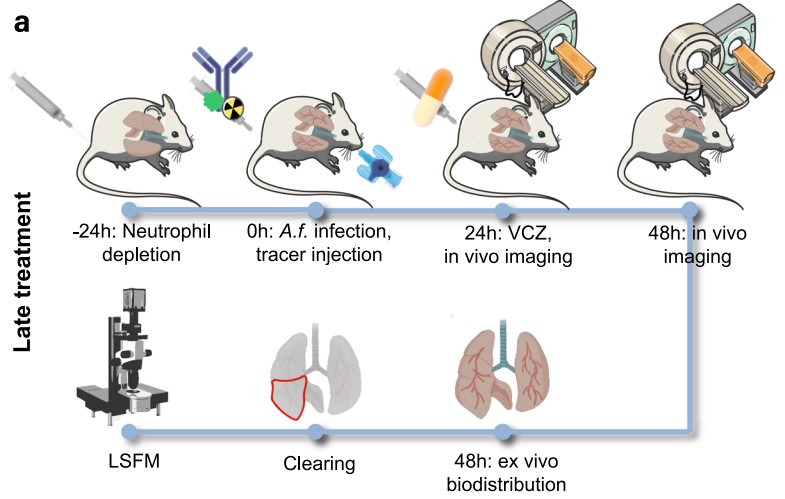

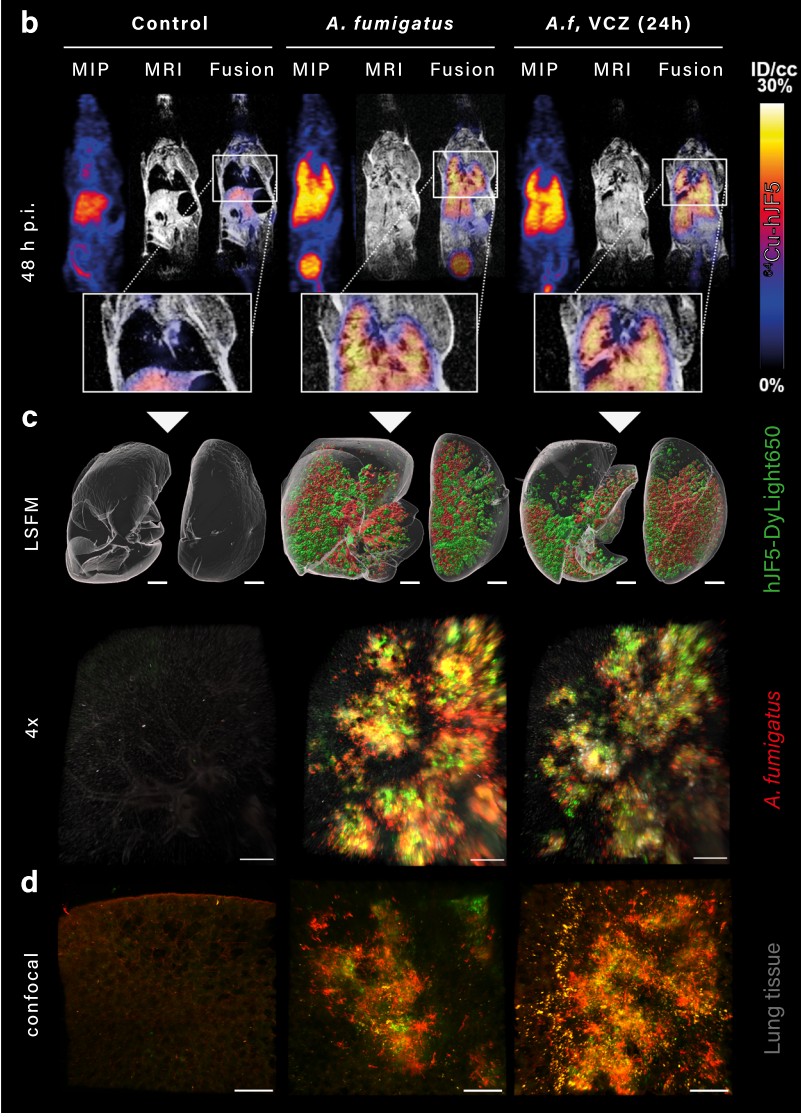

## Discussion

Early and accurate diagnosis and therapy monitoring of IPA remains a significant clinical challenge due to the paucity of tests that allow specific and sensitive detection of the disease[12]. The recent introductions of a novel CE-marked lateral-flow device[27,28] and a CE-marked ELISA test[29] that both use monoclonal antibody JF5 to detect a biomarker of *Aspergillus* infection have improved the speed and accuracy of IPA diagnosis, complementing the predicate galactomannan-ELISA test. Despite improved availability of biomarker tests over recent years, and significant efforts by the research community to develop non-invasive urine tests[30,31], diagnosis of IPA can be delayed for days,

**Fig. 3 Impact of antifungal treatment measured with PET/MRI and LSFM using dual-labeled $^{64}$Cu-hJF5-DyLight650. a** Experimental workflow depicting *Aspergillus fumigatus* (*A.f.*) infection, tracer injection, voriconazole administration (VCZ) and imaging using immunoPET/MR and LSFM. **b** ImmunoPET/MR images of PBS-deposited animals (Control, left, $n = 2$), *A. fumigatus*-infected animals without treatment (*A. fumigatus*, middle, $n = 2$) and animals with voriconazole treatment 24 h after inoculation (*A.f.* VCZ (24 h), right, $n = 3$). Images were acquired 48 h postinoculation (p.i.). MIP is a maximum intensity projection of the PET image; MRI is a T2-weighted MRI single slice MRI image; Fusion is a single slice PET image overlaid on the corresponding MRI slice. A close-up of the fusion image in the lung area is displayed below. PET data are presented as percent injected dose per cubic centimeter (%ID/cc). **c** Detection of $^{64}$Cu-hJF5-DyLight650 with LSFM. Surface view of lungs from same animals as in **b** imaged with LSFM, and with magnified detail shown below. Lung tissue (gray), *A. fumigatus* (red) and $^{64}$Cu-hJF5-DyLight650 signal (green). Scale bar first row = 1.5 mm, and second row = 300 µm. **d** High-resolution confocal images of same lungs as in **b**. Scale bar = 50 µm.

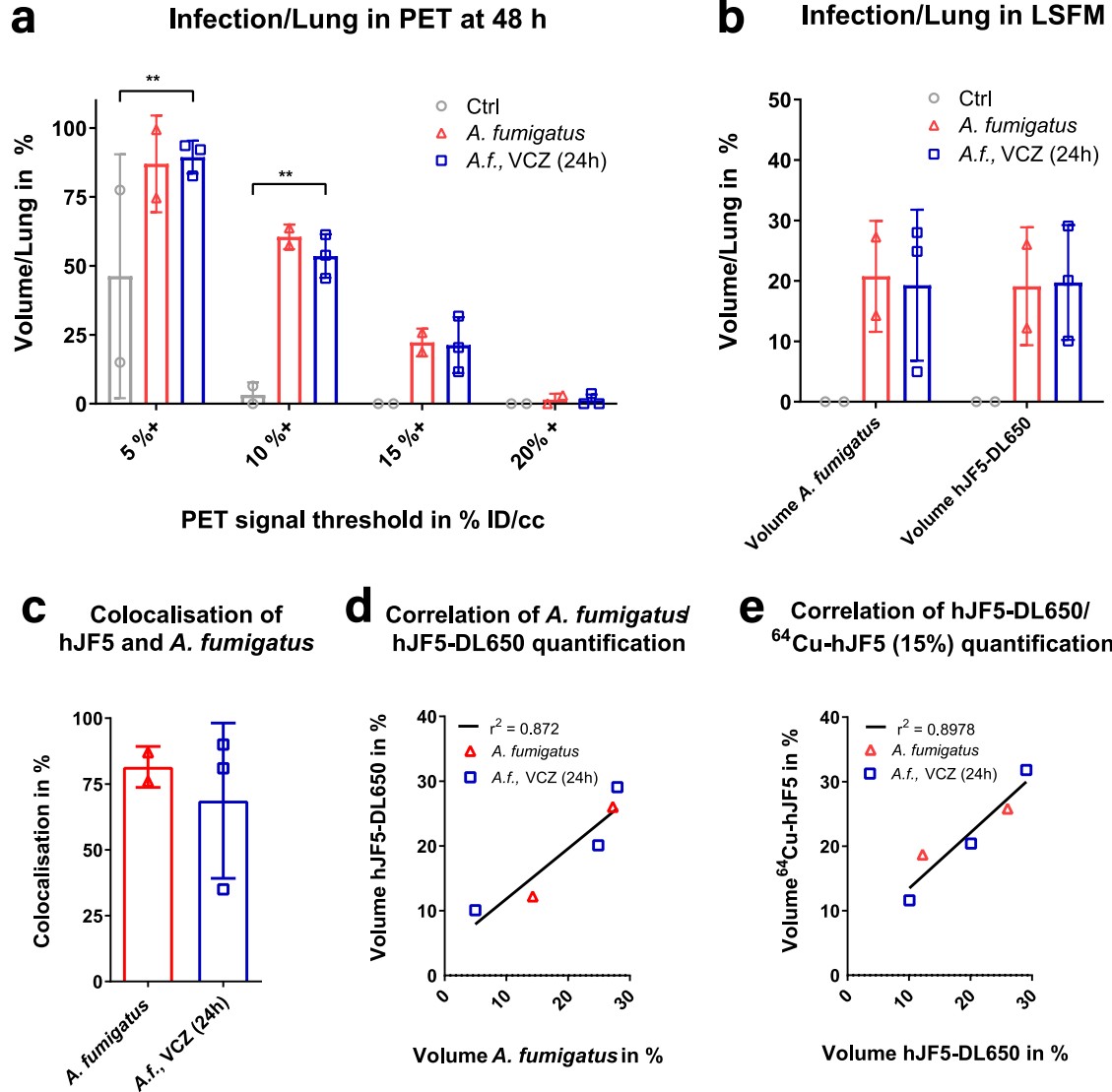

**Fig. 4 Quantification and volumetric analysis of immunoPET and LSFM. a** Ratio of infected lung volume as measured by immunoPET to total lung volume measured via MRI as a function of $^{64}$Cu-hJF5-DyLight650 accumulation threshold in percent injected dose per volume (%ID/cc) at 48 h postinoculation in PBS-deposited animals (Control, left, $n = 2$), *A. fumigatus*-infected animals without treatment (*A. fumigatus*, middle, $n = 2$) and animals with voriconazole treatment 24 h after inoculation (*A.f.* VCZ (24 h), right, $n = 3$). 5%: Ctrl vs. *A.f.*, VCZ (24 h), $p = 0.0062$. 10%: Ctrl vs. *A.f.*, VCZ (24 h), $p = 0.0013$. **b** Volumetric ratio of *A. fumigatus* or $^{64}$Cu-hJF5-DyLight650 in the lung as measured by LSFM for each group. **c** Quantification of the co-localization of $^{64}$Cu-hJF5-DyLight650 and *A. fumigatus*$^{tdTomato}$ measured by LSFM in untreated (*A. fumigatus*, red) and treated (*A.f.*, VCZ (24 h)). **d** Correlation of *A. fumigatus* invasion and $^{64}$Cu-hJF5-DyLight650 distribution in the lungs of *A. fumigatus*-infected animals untreated (*A. fumigatus*, red) and VCZ treated 24 h after infection (*A.f.*, VCZ (24 h) as measured by LSFM. **e** Correlation of $^{64}$Cu-hJF5-DyLight650 volumetric distribution in the lungs of *A. fumigatus*-infected animals untreated (*A. fumigatus*, red) and VCZ treated 24 h after infection (*A.f.*, VCZ (24 h) as measured in LSFM and PET using the 15%ID/cc threshold. $r^2$ represents Pearson's correlation coefficient. Results are plotted as individual values with average and standard deviation. $p$ values were generated using two way ANOVA with Tukey's multiple comparison test (*$p < 0.05$; **$p < 0.01$; ***$p < 0.001$; ****$p < 0.0001$).

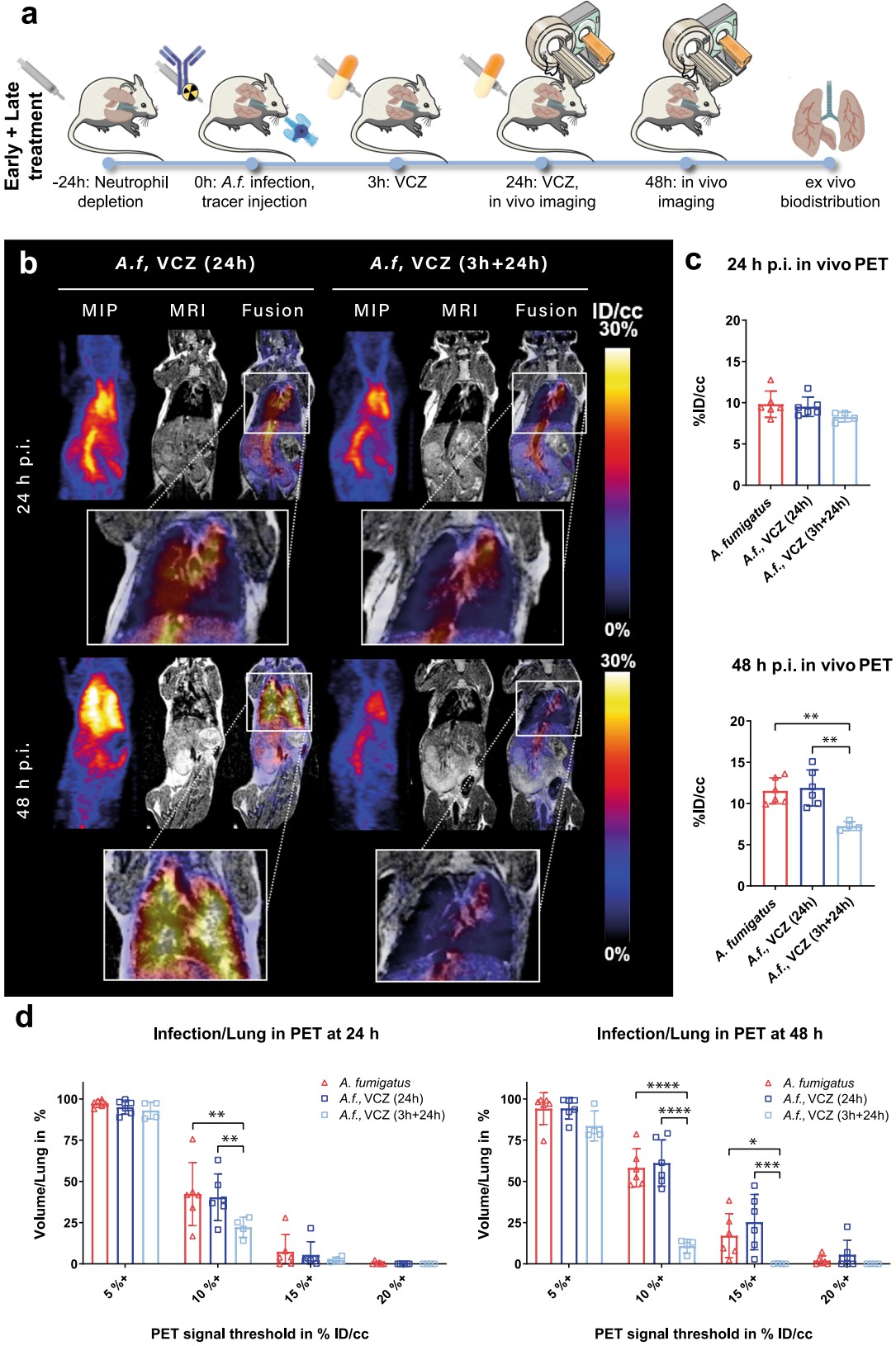

weeks or even months, due to outsourcing of testing to centralized facilities. Delays in test turnaround, and the need for timely treatment of febrile patients unresponsive to antibiotics, leads to the empiric use of antifungal drugs, and limits the use of biomarker tests as tools to monitor treatment efficacies. At present, no non-invasive diagnostic procedure for IPA exists that

provides sufficient specificity and sensitivity to allow for the quantitative detection of small amounts of fungus in situ, early diagnosis of *Aspergillus* infections in vivo, and response of the pathogen to antifungal drug treatment. To this end, the use of nuclear medicine techniques such as SPECT and PET imaging seem ideal, provided suitably specific radiotracers can be

**Fig. 5 ImmunoPET/MR evaluation of early- and late-stage voriconazole treatment on *A. fumigatus* lung infection. a** Experimental workflow depicting *Aspergillus fumigatus* (*A.f.*) infection, tracer injection, voriconazole administration (VCZ), and in vivo imaging using immunoPET/MR. **b** ImmunoPET/MR images of *A. fumigatus*-infected animals receiving voriconazole (VCZ) treatment initiated 24 h post inoculation (p.i.) (*A.f.* VCZ (24 h), left, *n* = 6), or when initiated at 3 h postinoculation and at 24 h p.i. (*A.f.* VCZ (3 + 24 h), right, *n* = 4). Images acquired 24 h p.i. are shown in the top row, while images in the bottom row show images acquired 48 h p.i.. MIP is a maximum intensity projection of the PET image; MRI is a T2-weighted MRI single slice MRI image; Fusion is a single slice PET image overlaid on the corresponding MRI slice. Magnifications of the fusion images of the lung are displayed below each image. **c** Quantification of $^{64}$Cu-hJF5 accumulation in the lung tissue at 24 h (top) and 48 h (bottom) p.i. of *A. fumigatus* in animals left untreated (*A. fumigatus*, red), receiving voriconazole treatment initiated 24 h after infection (*A.f.* VCZ (24 h), dark blue), or when initiated at 3 h postinfection and at 24 h postinfection (*A.f.* VCZ (3 + 24 h), light blue). 48 h p.i.: *A. fumigatus* vs. *A.f.*, VCZ (3 h + 24 h), *p* = 0.0046; *A.f.*, VCZ (24 h) vs. *A.f.*, VCZ (3 h + 24 h), *p* = 0.0025. **d** Percentage of lung volume occupied by thresholded PET signal at 24 h (left) and 48 h (right) postinfection in the same animals. At 24 h, 10%: *A. fumigatus* vs. *A.f.*, VCZ (3 h + 24 h), *p* = 0.0023; *A.f.*, VCZ (24 h) vs. *A.f.*, VCZ (3 h + 24 h), *p* = 0.0063. At 48 h, 10%: *A. fumigatus* vs. *A.f.*, VCZ (3 h + 24 h), *p* < 0.0001; *A.f.*, VCZ (24 h) vs. *A.f.*, VCZ (3 h + 24 h), *p* < 0.0001 and 15%: *A. fumigatus* vs. *A.f.*, VCZ (3 h + 24 h), *p* = 0.0321; *A.f.*, VCZ (24 h) vs. *A.f.*, VCZ (3 h + 24 h), *p* = 0.0009. Results are plotted as individual values with average and standard deviation and are expressed as percent injected dose per cubic centimeter (%ID/cc) for PET. *p* values were generated using one way ANOVA (**c**) or two way ANOVA (**d**) with Tukey's multiple comparison test (**p* < 0.05; ***p* < 0.01; ****p* < 0.001; *****p* < 0.0001).

developed that enable high affinity and selectivity for IPA. While recent studies suggest that [$^{18}$F]FDG-PET/CT might help discriminate between active and residual fungal lesions to support decisions for safely stopping antifungal treatments[32], we have shown that the metabolic tracer [$^{18}$F]FDG is not specific for IPA, and is unable to discriminate between *Aspergillus* infections, bacterial infections, and inflammatory processes of the lung[19]. Alternative strategies for imaging IPA are therefore needed. The three main alternatives investigated to date are (1) the use of radiolabeled antifungal drugs, which show poor ability to detect *A. fumigatus* infections, and to differentiate these from infections by other fungal pathogens[33,34]; (2) the use of radiolabeled siderophores[30,35,36] which, while demonstrating significant potential in animals models of disease, may lack sufficient specificity, and show reduced sensitivity in patients with iron overload (typically those at high risk of developing IPA including heavily transfused AML patients, neutropenic patients, liver and allogeneic HSCT recipients, and those receiving chemotherapy);[13] (3) the use of an *Aspergillus*-specific mAb, JF5, which allows highly specific and sensitive detection of *A. fumigatus* lung infections in vivo[13,19,20], and which has been humanized, via CDR grafting into a human IgG1 framework, for use as a diagnostic tracer in humans[20]. In parallel, optical detection of *A. fumigatus* has been shown in a fungal keratitis model using fluorescent liposomes, achieving rapid and sensitive diagnosis, although with an imaging depth too limited for lung imaging[37].

In this study, we extend our initial work to examine whether and how the hJF5 antibody can be used for early diagnosis and therapy monitoring of IPA. By coupling the antibody to the radionuclide $^{64}$Cu and the fluorophore DyLight 650, we have gained insight into the progression of *A. fumigatus* lung infections in vivo, and the response of infections to early- and late-stage treatment with the antifungal drug VCZ. Using light sheet fluorescence microscopy (LSFM), with a mesoscopic sample size similar to whole animal immunoPET, we were able to quantify, in 3D, uptake of the antibody tracer and development of a genetically modified strain of the pathogen (*A. fumigatus*$^{tdTomato}$), in parallel, in situ. We were therefore able to visualize growth of the fungus under different conditions in the intact lung both in vivo and ex vivo, including the role of the neutropenic environment on pathogen development, and the timing of therapy on disease progression. This demonstrates the utility of immunoPET as a nondestructive and noninvasive disease-monitoring tool to observe the in vivo response of *A. fumigatus* to antifungal drug treatment. Previous attempts to study the consequences of antifungal therapy on disease progression have relied on destructive, invasive, or semi-invasive procedures including histopathological examination of diseased lung tissues[22,38], culture of invasive BALf

samples for estimates of fungal load[7], and circulating biomarkers as proxies of disease burden[22,38–40]. It is important to mention that, while our efforts and model were based on the *Aspergillus fumigatus* species, the JF5 antibody recognizes and binds to all *Aspergillus* species and thus still holds potential in parts of the world where *A. flavus* or *A. terreus* can be the most common cause of IPA[18]. In addition, we have focused on the lung presentation of the disease. While rare, cerebral aspergillosis can also occur, especially in patients treated with ibrutinib. Here, our chosen antibody-based approach might be limited as it would require breakdown of the blood brain barrier (BBB) to allow for radiolabeled antibody accumulation at the site of infection. However, *A. fumigatus* metabolites such as gliotoxins[41] have been shown to impair BBB integrity.

Neutrophils play critical roles in host defence to fungal infections[42]. In our neutropenic mouse model of IPA, neutrophil deficiency, more specifically loss of Gr-1$^+$ cells, resulted in infection of the entire lung volume within a matter of hours of challenge with spore inoculum, whereas, in the presence of neutrophils, the pathogen was rapidly controlled. This demonstrates further the critical role of Gr-1$^+$ neutrophils in preventing infectious pathogens[43], and provides an additional, and highly effective, means of depleting neutrophils, to those used by Amich and co-workers in LSFM of *A. fumigatus* lung infections[44]. Surprisingly, in healthy animals, neutrophil-mediated control of lung infection took longer than expected, as invasion was apparent in LSFM 6 h after initial deposition of *A. fumigatus* spores. However, the spore concentrations used in our investigations were superficially high in order to obtain a complete and rapid invasion of the lung, and were several orders of magnitude higher than the daily estimated inhalation of *A. fumigatus* spores in real-life[45]. Nevertheless, the timing of the different stages of lung infection (conidial germination, hyphal proliferation, and invasive of the contiguous lung) are similar to those reported in other neutropenic models of IPA[22,38].

In order to determine the effectiveness of our molecular imaging approach as a means of monitoring the in vivo response of invasive lung infection to antifungal treatment, we administered the azole drug VCZ at early (3 h) and late (24 h) time points postinoculation, and then quantified spatio-temporal changes in pathogen load using immunoPET. In PET imaging, no effect of the drug was found on the distribution and accumulation of $^{64}$Cu-hJF5 in the lungs or other organs in infected animals, when administered 24 h postinoculation. Even at 48 h postinoculation, 24 h after the drug was administered, no control of the disease was evident. A possible explanation for this might be the biological activity of VCZ, "holding" *A. fumigatus*[46] infections by a fungistatic effect until the immune system can take over. While

the effect of VCZ in limited time frames is, to the best of our knowledge, undocumented in vivo, its initial activity in vitro is likely to be fungistatic, with fungicidal activity following 24–48 h earliest after initial fungal exposure[47–49]. Data from LSFM of the lungs confirmed the immunoPET data, and similarly showed a lack of reduction in IPA following late-stage administration of VCZ. Our data showed that, in order to be effective, VCZ must be administered as early as possible during the infection process in the setting of neutropenia, with early/late-stage treatment providing complete remission of the disease. Our findings are consistent with previous studies which have shown that early intervention with antifungal drugs is critical to control of IPA[38]. Our work has also demonstrated the power of JF5-guided imaging as a means of tracking IPA in vivo during the course of antifungal drug treatment. As a highly sensitive and disease-specific tracer, and one that detects active growth of *A. fumigatus*[13,18–20], ⁶⁴Cu-hJF5 represents a manifest improvement over [¹⁸F]FDG-PET/CT[32] as a means of discriminating between invasive growth and resolution of infection. Accumulation of the hJF5 tracer was similar in both treated and non-treated animals, and correlated exclusively with the distribution of the pathogen. This demonstrates the absence of false positive accumulation (hJF5 accumulating in *A. fumigatus*-absent spaces) and false negativity (hJF5 not accumulating in *A. fumigatus*-rich spaces). In addition, the ex vivo results showed a discrepancy between radiotracer uptake in the spleen of animals infected with *A. fumigatus* compared to animals with a sham inoculation. This might point toward dissemination of the pathogen to the spleen, which has been reported several times in humans[50,51]. However, it might alternatively demonstrate accretion of soluble antigen released into the bloodstream during invasion of the lung.

While the extended circulation time of full-length antibodies (days to weeks) and their high bio-availability are a significant advantage in many instances, these features do present an important limitation for immunoPET in vivo, which is a high blood pool of tracer. In our system, this typically manifested as blood quantities of >20%ID/g in ex vivo biodistribution assays and, more importantly, in vivo accumulation of ⁶⁴Cu-hJF5 in the highly perfused lungs of noninfected control animals. Biodistribution studies showed a difference in blood circulation values between infected (treated and non-treated) and control animals, which we attribute to circulating antigen. We believe this PET signal still originates from immunoreactive radiolabeled antibody, as we have shown in previous studies the stability of ⁶⁴Cu chelation to NODAGA-hJF5 as well as the conjugation between NODAGA and hJF5[20]. Since it is not possible to distinguish blood vessels from lung tissue in immunoPET/MR, this translates to a background signal that reduces the contrast between healthy and diseased tissues in in vivo PET/MR. To mitigate this, we conducted ex vivo measurements after thorough perfusion of the animals with PBS in order to remove any unbound circulating ⁶⁴Cu-hJF5 thereby allowing the distinction between circulating radiotracer and bound immunoglobulin. The benefits of this approach were clearly evident in LSFM, where control animals showed no fluorescence signal at all due to ⁶⁴Cu-hJF5-DyLight650. Furthermore, the accumulation of ⁶⁴Cu-hJF5 determined in ex vivo biodistribution assays was ~10-fold lower in control animals compared to infected animals, and an approximate 2-fold difference was achieved in in vivo PET at the same time point. While this is a sizeable difference, it is very likely a feature unique to the murine model of IPA since the lungs are fully invaded by small nodules of fungus, which is apparent in the LSFM images. This results in a dispersed mass at the resolution of the PET system (~1 mm). This might lead to a partial volume effect, ultimately lowering the estimated PET accumulation

values. In the case of a naturally infected human, IPA generally presents as several compact nodules of a few centimeters against a healthy tissue background[52]. Given the organ and nodule size in relation to the conserved PET resolution, the contrast ratio between positive nodules and healthy lung tissue should increase considerably, and prove sufficiently different, for human disease diagnosis and therapy monitoring. A possible means of reducing the blood pool retention of the radiotracer, is to shorten the circulation time of the targeting vector through chemical manipulation. Antibody fragments are a classical approach, as smaller fragments often conserve the affinity to the target of interest but with a more limited circulation time, with typical half-life times of 10 h for F(ab')₂ fragments or even less than 1 h for nanobodies[53]. Such strategies are currently being investigated by our group to achieve a higher contrast ratio between diseased and healthy tissue at a faster rate than currently achievable with a full-length radiolabeled antibody.

Notwithstanding this, and in order to achieve a higher contrast ratio in vivo as well as better nodule delineation and disease staging, we developed a volumetric thresholding method applicable to both LSFM and PET images. This allowed the values acquired in vivo and ex vivo to be correlated, and provided a more robust mechanism for disease detection. To achieve this, a delineation of the organ was performed using a reference anatomical image (MRI or autofluorescence for immunoPET and LSFM, respectively), and the volume occupied by the specific signal above a defined threshold value calculated. Initially, we used a threshold value similar to that of the average background levels (5%ID/cc), which yielded too much noise and variance to allow conclusive results. While using a 10%ID/cc threshold provided more meaningful results, the volumes occupied by the pathogen were largely over-estimated when compared to the results obtained by LSFM. This could be due to the high perfusion of the lung (and blood containing ~20%ID/g), giving false positive signals indistinguishable from true nodule signals achievable at the resolution of small animal PET. By increasing the threshold to 15%ID/cc, representing the maximal PET signal in healthy animal lungs, we were able to obtain a clear delineation of the diseased regions of the lungs, often matching with MRI T2 contrast, with values similar to the LSFM values obtained ex vivo. Furthermore, we achieved a significantly higher ratio between diseased and control lungs in vivo than previously achieved using classical PET quantification approaches. This effectively solved the main immunoPET issue of high blood pool retention of mAb tracer.

In conclusion, we have shown that hJF5 can be used successfully in vivo as a diagnostic and therapy monitoring tool for IPA, accurately tracking changes in pathogen load in the neutropenic lung during disease progression and in response to antifungal drug treatment. It is important to note that while this murine model of IPA is an accurate representation of the clinical aggressiveness of the pathogen in the context of immuno-suppression, clinical presentation of IPA is typically as discrete nodules rather than diffuse invasion of the contiguous lung. In this event, the volumetric threshold procedure developed here might prove useful for clinical diagnosis of the disease using antibody-guided molecular imaging. We foresee that the main applications of this immunoPET approach could be firstly to rule-in or rule-out *Aspergillus* lung infection, and secondly to aid identification of treatment end-point, improving clinical decisions regarding duration of the antifungal therapy.

## Methods

**Preparation of labeled hJF5.** For conjugation of hJF5 to NODAGA, purified hJF5 antibody was first incubated with EDTA (0.5 mM) for 45 min at room temperature under gentle stirring to remove metal contamination. Buffer exchange and EDTA excess removal was then performed using trace-metal basis 0.1 M boric acid

(Sigma-Aldrich, Switzerland), pH 9.1, equilibrated with PD-10 (GE Healthcare, Switzerland). Amicon Ultra-15 centrifugal filter units with a 10 kDa molecular weight cut-off (Merck Millipore, Switzerland) were then used for antibody collection. p-NCS-Bz-NODAGA (Chematech, France) was dissolved in DMSO at 33.3 mM and applied at a 20-fold molecular excess to the antibody solution (33.3 µM) and the reaction took place for 16 h at 4 °C. After purification, performed with a PD-10 column, the antibody was equilibrated in Chelex-treated PBS, pH 7.4 (Bio-Rad, Switzerland). Filtration using Amicon Ultra-15 centrifugal filter units (10 kDa cut off) and two further purification steps using Chelex-treated PBS (pH 7.4) followed[19].

For dual-labeled hJF5, the chelator-conjugated antibody was further conjugated with a 10-fold molar excess of DyLight 650 NHS ester (Thermo Scientific, Rockford, IL, USA) according to the manufacturer's instructions. The conjugate was purified using a PD-10 column equilibrated with phosphate-buffered saline. The antibody was radiolabeled by incubation with 0.5 MBq $^{64}$Cu per µg for 60 min at 42 °C.

$^{64}$Cu was produced in house by 12.5 MeV proton irradiation of enriched $^{64}$Ni metal electroplated on a platinum/iridium disc via the $^{64}$Ni(p,n)$^{64}$Cu nuclear reaction, and purified after acidic dissolution by anion exchange chromatography. Antibodies coupled to NODAGA were then radiolabeled using 0.5 MBq of acetate-buffered $^{64}$CuCl$_2$ per µg of antibody.

Quality control was performed using thin layer chromatography (Polygram SIL G/UV254, Macherey-Nagel; mobile phase: 0.1 M sodium citrate, pH 5.0) and HPSEC (Phenomenex BioSep SEC-s3000 300 × 4.6 mm, 1.5 ml/min saline sodium citrate). Reactivity of antibody conjugates was confirmed by ELISA with purified antigen as described previously[20].

**Aspergillus fumigatus strains and voriconazole susceptibility testing**. The *Aspergillus fumigatus* wild-type strain ATCC 46645 and strain *A. fumigatus*$^{tdTomato}$ (strain ATCC 46645 with insertion of the tdTomato gene via pSK379-Tomato plasmid in the His2 sequence[54]) were grown on defined minimal medium for three days at 37 °C before resting conidia were harvested from sporulating mycelium with tap water + 0.1% Tween-20. The growth medium of the *A. fumigatus*$^{tdTomato}$ strain was supplemented with 0.1 µg/ml pyrithiamine hydrobromide (Sigma-Aldrich, St. Louis, Missouri, USA) as selection marker. Spore suspensions were filtered through a series of cell strainers with decreasing mesh size (100, 30, and 10 µm) before spore concentration was assessed using an automated cell counting system (Cellometer Auto T4; Nexcelcom, Manchester, UK). After adjusting the spore concentration to 3 × 10$^5$ spores/ml with deionized water, 100 µl of these suspensions were used per well to inoculate the EUCAST 96-well flat-bottom testing plates (TPP, Trasadingen, Switzerland). Previously, these standard cell culture plates were filled with 100 µl double-concentrated EUCAST medium (RPMI 1640 (with glutamine and a pH indicator but without bicarbonate) + [4%] (w/v) Glucose + [0.33 mol/l] 3-(N-morpholino) propanesulfonic acid (MOPS)) per well with which double-concentrated dilution series of voriconazole (Vfend; Pfizer, New York City, USA; reconstituted with sterile, deionized water) were generated (32, 16, 8, 4, 2, 1, 0.5, 0.25, 0.125, 0.06 µg/ml). By adding 100 µl of the spore suspensions to the wells of the 96-well plates the total spore number per well equated to 3 × 10$^4$, the EUCAST medium was diluted to become 1× concentrated (RPMI 1640 + [2%] Glucose + [0.165 mol/l] MOPS), and the final voriconazole concentrations were adjusted to 16, 8, 4, 2, 1, 0.5, 0.25, 0.125, 0.06, 0.03 µg/ml. For each strain, two individual spore preparations were used (=biological replicates) from which three separate voriconazole dilution series were inoculated (=technical replicates). The inoculated plates were then incubated for 48 h at 37 °C before the MIC values were determined. For visual MIC determination, all inoculated testing rows were investigated with a stereo microscope (S8 APO; Leica Microsystems, Wetzlar, Germany) for the first well of the row in which no fungal growth was detectable. Subsequently, for assessing the MIC values spectrophotometrically, the absorbance at 490 nm of all inoculated wells of the 96-well plates was measured (iMark; Biorad, Hercules, California, USA). As blank control, fresh 1× concentrated EUCAST medium was used.

**Mouse strain**. For all experiments, C57BL/6JOlaHsd mice were used. Eight to ten-week-old female mice were purchased from Envigo (Huntingdon, Cambridgeshire, UK), and kept in a standardized and sterile environment (20 ± 1 °C room temperature, 50 ± 10% relative humidity, 12 h light–dark cycle), with food and water provided ad libitum. All experiments were undertaken in accordance to the animal welfare act and the EU directive 2010/63/EU, the German animal welfare act and after review and approval by the responsible local ethical committees and authorities (Regierungspräsidium Tübingen and Essen, Germany).

**A. fumigatus infection and treatment**. Mice were rendered neutropenic by an intraperitoneal (i.p.) injection of 100–200 µg anti-Ly6G/anti-Ly6c antibody (BioXCell, West Lebanon, USA) diluted to 1 mg/ml in PBS (DPBS, Gibco Life technologies, Carlsbad, CA, USA) 18–24 h prior to fungal infection or injected by an equivalent amount of isotype antibody for the isotype-depleted group. For the infection procedure, mice were first anaesthetized with 100 µl of Ketamin/Xylazin or Rompun solution i.p. (Ketamin: 80–100 mg/kg. Ratiopharm GmbH, Ulm; Rompun: 15 mg/kg. Bayer HeathCare, Leverkusen or Xylazin 2%: 10 mg/kg. WDT,

Garbsen), before intubation using a 22-gauge in-dwelling venous catheter (Vasofix Braunüle, B. Braun AG, Melsungen, Germany). Correct intubation in the trachea was confirmed by ventilation, before deposition of a suspension of 4–5 × 10$^6$ spores of *A. fumigatus* in 50–100 µl PBS (MiniVent, Hugo Sachs, March-Hugstetten, Germany). Either wild-type *A. fumigatus* or *A. fumigatus*$^{tdTomato}$ were used depending on the experiment. Complete distribution in the lungs was obtained by artificial ventilation (250–300 µl/breath at 200–250 breaths/min for 1 min using a small animal respirator). Control animals received an equal volume of the PBS carrier only.

For immunoPET/MRI experiments, mice were then randomly split into groups, and received the following treatments: (1) no antifungal treatment (*A. fumigatus* only group); (2) a single i.v. injection of 6 mg/kg of VCZ (Hexal, Holzkirchen, Germany) 24 h after infection (late-stage VCZ treatment); (3) an i.v. injection of 6 mg/kg of VCZ 3 h after infection, followed by a second injection of 4 mg/kg of VCZ 24 h after infection (early/late-stage VCZ treatment).

Immediately after infection, mice were separated by experimental groups to avoid cross contamination, in litters of maximum 6 littermates. Thus, no blinding of the investigator was possible.

**In vivo immunoPET/MR imaging**. Directly after intratracheal infection of the animals, ~12.5 MBq of radiolabeled hJF5, corresponding to ~25 µg of protein ($^{64}$Cu-hJF5 injected 3.04 mg/ml, $^{64}$Cu-hJF5-DyLight650 at 1.25 mg/ml) at, was injected i.v. In vivo imaging took place 24 and 48 h after initial infection of the animals using a small animal PET system (Inveon™, Siemens Healthcare GmbH, Erlangen, Germany), using the Inveon Acquisition Workplace (IAW) v2.1.272 software (Siemens Preclinical Solutions, Knoxville, TN, USA) thoroughly calibrated[55] and undergoing daily quality and linearity response controls, ensuring reproducibility of measurements, and a small animal MRI using the Paravision v6 software (7 T Bruker Biospin GmbH, Billerica, MA, USA). Before each imaging session, the animals were weighed and their appearance was checked for physical or behavioral abnormalities. During the scans, the animals were anaesthetized with 0.8–2.5% isoflurane in 100% oxygen (depending on their health and fitness). Anesthesia was monitored by measuring the breathing frequency. Mice were positioned into a PET and MRI transparent animal holder alongside two capillaries containing a minimal amount of radioactivity for coregistration of PET and MRI imaging data. For acquisition, a 600 s static PET acquisition sequence was used followed by two MRI sequences (T2_TurboRARE (TR/TE 600 ms/35 ms, size 256 × 128 × 92 mm, slice thickness 23 mm, field of view (FOV) 64 × 32 × 12 mm, and resolution 0.25 × 0.25 × 0.25) and T1-FLASH (TR/TE 12/3 ms, FA 8.9°, size 160 × 80 × 60 mm, slice thickness 18.868 mm, FOV 50 × 25 × 18.868 mm, and resolution 0.314 × 0.314 × 0.314)). PET image reconstruction was performed using a 3D ordered subset expectation maximization (3D-OSEM) with no attenuation correction in IAW and co-registered with MRI images using in Inveon Research Workplace software v4.2 (Siemens Preclinical Solutions, Knoxville, TN, USA). Region of interest (ROI) was drawn onto the lungs using the MR Images for guidance, and the PET data were used to extract percent injected dose per cubic centimeter (cc) [%ID/cc] in the lung. Lung volumes were measured using the ROI as delineated in the MRI, or measured based on the PET data after thresholding the initial lung ROI at a defined %ID/cc value.

**Ex vivo analysis**. After the last in vivo imaging scans, retro-bulbar blood samples were obtained from the animals before CO$_2$ sacrifice. Shortly thereafter, all animals were perfused through the left ventricle with 0.4% of PFA (Sigma-Aldrich, Darmstadt, Germany) in 20 ml of 4 °C PBS. Afterwards, organs were removed and radioactivity was quantified with an aliquot of the injected radiotracer in a γ-counter (Wallac 2480 WIZARD 3″, PerkinElmer, Waltham, MA, USA) using an energy window between 350 and 650 keV. The results are expressed as % injected dose per gram (%ID/g) of tissue.

**CUBIC clearing and 3D fluorescence microscopy**. Organs prepared for 3D fluorescence microscopy were cleared with the CUBIC protocol as previously described[56]. All incubation steps were performed in the dark. Fixated lungs were transferred to 0.01% NaN$_3$/PBS (Sigma-Aldrich, Darmstadt, Germany) overnight at RT with shaking. After an additional washing step with 0.01% NaN$_3$/PBS, the lungs were incubated in 50% CUBIC reagent-1/dH$_2$O and 3 h at RT with shaking. Reagent-1 was prepared according to the published protocol[56] and was comprised of 25% Urea (Sigma-Aldrich, Darmstadt, Germany), 25% Quadrol (Sigma-Aldrich, Darmstadt, Germany) and 15% Triton X-100 (Sigma-Aldrich, Darmstadt, Germany) in dH2O. On the next day, the samples were transferred in 100% reagent-1 over night at RT with continuous shaking, followed by replacement of the solvent by fresh reagent-1 on the next day for an additional 24 h period incubation. Following the incubation period, 3 × 2 h washing cycles at RT with 0.01% NaN3/PBS were performed with shaking and the lungs immediately immersed in 50% CUBIC reagent-1/PBS over night at RT. Reagent-2 was prepared following the established protocol[56] with 25% urea, 50% sucrose (Sigma-Aldrich), and 10% triethanolamine (Sigma-Aldrich) in dH$_2$O. Final clearing was performed in pure reagent-2 overnight at RT with gentle shaking, followed by a last change of reagent-2 for at least 24 h at RT, with shaking. Before image acquisition, samples were preincubated in a

mixture of 50% silicon oil (Sigma-Aldrich) and 50% mineral oil (Sigma-Aldrich) 30 min, before imaging in the same oil composition.

All samples were imaged using an Ultramicroscope II and ImSpector software (LaVision BioTec, Bielefeld, Germany). The microscope is based on a MVX10 zoom body (Olympus, Hamburg, Deutschland) with a ×2 dipping objective, and equipped with a Neo sCMOS camera (Andor, Oxford instruments, Tubney Woods, Abingdon, UK). Samples were imaged along the coronal axis in a commercially available sample holder and cuvette (LaVision BioTec), with a total magnification of ×1.26 or ×4 and a z spacing of 10 or 5 μm between optical planes. A FOV with $2560 \times 2160$ pixel and a diagonal of 17.6 or 5.5 mm resulted in a x/y pixel size of 5.16 or 1.62 μm. For image acquisition, samples were excited with light sheets of the wavelengths 488, 561, and 639 nm. The longest wavelength was imaged first and shortest last to avoid photobleaching during imaging. A bandpass emission filter with a center wavelength of 525 nm and a bandpass width of 50 nm (525/50) was used for the detection of autofluorescence, 585/40 for the detection of the tdTomato signal of *A. fumigatus* and 680/30 for the DyLight650 signal of the hJF5 antibody.

Multiphoton images of fungal development were generated from the right inferior lobes using the Leica TCS SP8 MP and Leica Application Suite (LAS X) software (Version 3.1.5.16308; Leica Microsystems GmbH, Wetzlar, Germany). Z-stacks of 150 μm with a spacing of 0.5 μm were acquired using the 2.8 W Chameleon Vision II Titan-Sapphire laser with an excitation wavelength of 960 nm. A ×25 objective (Leica Microsystems GmbH) with a ×1.5 zoom and a scan speed of 100 Hz resulted in a FOV with the size of 295.24 μm × 295.24 μm in $1024 \times 1024$ pixels, and a x/y pixel size of 0.289 μm. Autofluorescence was detected with a photomultiplier tube (PMT) bandpass emission filter 525/50, and tdTomato signal with a hybrid detector (HyD) 585/40.

Confocal images of the whole lungs, first imaged using PET/MRI, were acquired using the Leica TCS SP8 gSTED equipped with a 1.5 mW tuneable white light laser and LAS X software (Version 3.5.5.19976; Leica Microsystems GmbH). With a ×20 objective (Leica Microsystems GmbH), excitation wavelengths of 488 nm (52%), 554 nm (63%), 652 nm (63%) and a scan speed of 400 Hz, z-stacks of around 200 μm with a spacing of 1 μm were acquired. A FOV with the size of 581.25 μm × 581.25 μm ($2048 \times 20{,}148$ pixels) results in a x/y pixel size of 0.284 μm. Autofluorescence was detected with a tuneable PMT of 499–525 nm, tdTomato signal with a tuneable HyD of 561–626 nm, and DyLight650 with an HyD of 670–718 nm.

**Image processing**. Recorded z-stacks (16 bit OME.TIF) were converted into Imaris files (.ims (Bitplane AG, Oxford instruments, Zurich, Switzerland)) using the ImarisFileConverterx64 (Version 9.4.X and 9.5.X). The 3D reconstruction of the surfaces, and the subsequent analyses, were performed with Imaris (Version 9.5.X, Bitplane). The displayed surface of *A. fumigatus*tdTomato signal was built using a threshold from 1000 to 3969.38 counts, and the displayed surface of hJF5-DyLight650 signal with 500 to 1117.71 counts. For the co-localization of both signals, we reconstructed a new surface from the *A. fumigatus* surface with the threshold values from the hJF5 surface reconstruction and calculated the overlap.

**Histology**. After sacrifice, animals were perfused with 20 ml PBS at 4 °C before organ removal. Whole lungs were fixed in 4% formalin and paraffin embedded. Tissue sections of 3–5 μm thick were prepared and stained with H&E and Grocott to demonstrate fungal structures. Appropriate positive and negative controls were used to confirm the adequacy of the staining. Photomicrographic images were acquired with an Axioskop 2 plus Zeiss microscope equipped with a Jenoptik (Laser Optik System, Jena, Germany) ProgRes C10 plus camera and software. Final image preparation was performed with Adobe Photoshop CS6.

**Statistical analysis**. Sample sizes were chosen based on previously published studies using similar conditions[20]. Infected animals were removed from the study based on previously established criteria of weight loss, which had to be between 5% and 20% between the day of infection and the 48 h timepoint, indicating a failed infection or requiring ethical sacrifice, respectively. Statistical analysis was performed using GraphPad Prism 9.0.0 for Windows (GraphPad Software, San Diego, CA, USA). Given data are indicated in the figure legends. Specific tests to determine statistical significance are noted in each sure legend. Correlation was acquired with linear regression best-fit, and the Pearson's correlation coefficient r was calculated. p values are depicted, with a value of $p < 0.05$ considered significant. Collected PET and Biodistribution data passed the Shapiro–Wilk's normality test.

**Reporting summary**. Further information on research design is available in the Nature Research Reporting Summary linked to this article.

## Data availability

Due to their large size, the raw imaging data that support the findings of this study are directly available from the corresponding authors upon reasonable request. Derived data have been compiled in the Source Data file provided with this paper. Any remaining data supporting the findings from this study are available from the corresponding author upon reasonable request.

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

## Acknowledgements

We are grateful to Walter Ehrlichmann and Dominik Seyfried for the radiotracer production. We thank Maren Harant, Linda Schramm, and Nathalie Mucha for their expert technical assistance, Simon Freisinger for his experimental help, Julia Mannheim, Johannes Schwenck, Stefan Wiehr, and Anna-Maria Wild for their scientific input, and the Imaging Center Essen (IMCES) for supporting all optical imaging. We thank Sven Krappmann for providing the A. fumigatus^tdTomato strain. We acknowledge funding by the Ministry of Culture and Science of North Rhine-Westphalia, the Governing Mayor of Berlin including Science and Research, and the Federal Ministry of Education and Research to M.G. This work was supported by the European Union Seventh Framework Programme FP7/2007–2013 under Grant 602820, and the Werner Siemens Foundation.

## Author contributions

Conceptualization and planning: S.H., M.H., C.R.T., B.J.P., M.G., and N.B.; Experimentation: S.H., A.M., L.B., F.N., I.G.M., A.K., M.H., and N.B.; Data analysis: S.H., and N.B.; Manuscript and figures preparation: S.H., A.H., I.G.M., A.K., M.H., C.R.T., B.J.P., M.G., and N.B.

## Funding

## Competing interests

C.R.T. is director of ISCA Diagnostics Ltd. The remaining authors declare no competing interests.
