## [Peer Review File · Nature Communications]

REVIEWER COMMENTS

Reviewer #1 (Remarks to the Author):

The authors have developed an exemplary imaging system for monitoring experimental invasive aspergillosis. They have shown the distribution of infection in the lung after experimental infection, and its progression in a short time frame in neutropenic mice. They contrast direct visualization of the fungus itself with a fluorophore-linked monoclonal antibody, that is used commercially for diagnosis. Over a short time frame (24 hours) they show the impact of voriconazole.

Comments

Methods

1. L401 The names or numbers of the *A. fumigatus* strains used need to be stated.
2. Do the authors know that both strains of *Aspergillus* that they used were susceptible to voriconazole? There refer to 'wild type', which was clearly pathogenic, but many strains in the environment and from patients in Europe are resistant to azoles. What are the MICs? And have they assessed fungicidal activity for these strains in vitro (the MFCs)? About 60% of *A. fumigatus* strains are killed by voriconazole in vitro (Warn et al, PMID: 16284102). This needs to be done.

General

3. Abstract and L97 – the phraseology of 'early intervention' to 'prevent' makes little sense. If infection is established, it cannot be prevented.
4. L242. The GM ELISA test was introduced 20 years ago. What is new are the lateral flow assays.
5. L251 Re other non-invasive tests – consider also Skriba et al (PMID: 30349512) and Marr et al (PMID: 29684106) as new (not yet in the clinic) alternatives, that should be mentioned.
6. L279 Consider fungal keratitis detection Lee et al (PMID: 29228372), although not used to follow treatment.
7. L304 Not correct. Needs rephrasing. In general, voriconazole 'holds' infection in profound neutropenia, until neutrophil recovery occurs. It is not expected that it would eradicate (or even lessen) infection in this short time frame with some help from the immune system. Some isolates are killed by voriconazole, others not.

8. In terms of clinical utility, is the diagnosis of *Aspergillus* nodules in cases where lung cancer is the main differential diagnosis another possible clinical case? See Baxter et al, PMID: 21460371 and Muldoon et al, PMID: 27538521). The authors mention cases of IA in severe influenza and COVID-19, but it is not realistic to imagine that such severely ill patients could be taken for PET imaging – another limitation. One of the other challenges clinically is determining when to stop therapy as CT and CXR imaging lags behind clinical cure. Could this technology allow the end point of antifungal therapy to be determined, shortening treatment periods and allowing additional immunosuppressive procedures to be done earlier with greater chances of success?

9. Not all cases of IA are caused by *A. fumigatus*. The species specificity of the J5 antibody should be stated and if not covering the common infecting species, mentioned as a limitation. In many parts of the world, *A. flavus* is more common than *A. fumigatus* and in some institutions *A. terreus* is particularly common.

10. A single sentence comment on brain imaging based on the blood brain barrier penetration of their reagents would be helpful. In irbutinib-treated patients, cerebral aspergillosis is much more common proportionately.

David Denning

Reviewer #2 (Remarks to the Author):

Hennenberg et al describe application of in vivo antibody-based imaging to the detection of *Aspergillus fumigatus* infections, evaluation of extent of disease, and response to treatment. There is no question that aspergillus infections remain an important clinical challenge, particularly in patients with impaired immunity, lung disease, or as an aftermath of viral illnesses. It is rewarding to see molecular imaging approaches increasingly and effectively being applied in infectious disease. Here, a humanized *Aspergillus*-specific antibody, JF5, has been dually-labeled (with Cu-64 and DyLight 650) for PET and fluorescence imaging in an aggressive mouse model of disease. The approach is further applied to assess response to treatment with voriconazole (VCZ).

The imaging studies are well executed and supported by corroborative studies. Strengths of the work include validation of the imaging observations via ex vivo microscopy including vivid 3D light-sheet microscopy and confocal microscopy. Furthermore, since the *A. fumigatus* strain used expresses a red fluorescent protein (Tomato), LSFM was used to confirm colocalization of the DyLight-labeled antibody in the mouse lungs. The investigators approach quantitation rigorously, exploring different cut-offs to attempt to establish objective criteria for confirming disease.

One of the challenges of the current work is the use of a severe model: mice which have been rendered neutropenic, allowing very rapid proliferation of the fungus. Normal mice resolve disease over 6-24 h, but the neutropenic mice develop persistent, fulminant infection over the 6-48 h time course. Treatment at 24 h is ineffective, but two doses, at 3 + 24 h appears effective. Conclusions drawn would not necessarily be relevant or even feasible in the clinical setting (i.e. starting treatment within 3 h of infection; patients are not infected at a specific time).

A significant challenge is quantification of the extent of fungal invasion. The authors take a systematic approach, establishing a threshold of activity in the PET scans that allows them to distinguish healthy vs diseased animals. Radiotracer activity levels of 5%ID/g did not differentiate between healthy and sick; 10% ID/g did identify diseased animals, but a fairly high threshold of 15% ID/g was needed to eliminate a false positive signal in control mice. Using this threshold (15%ID/g), an infection volume could be defined. The necessity of a high threshold can potentially reduce sensitivity. The fundamental problem is the use of an intact antibody with extended persistence in the circulation, leading to high activity in highly vascularized lung tissue (due to blood activity ~20%ID/g). The authors should discuss alternatives such as smaller antibody fragments or small protein scaffolds with more rapid elimination from the circulation. Also, this quantitation scheme is highly specific to the neutropenic mouse model and would need to be re-established for each new model, or for clinical use.

Nonetheless, the ability to visualize disease progression and optimize treatment through non-invasive imaging demonstrates the potential impact of this approach in infectious disease. The use of a pathogen-specific immunoPET probe clearly provides advantages over more general tracers such as ¹⁸F-FDG. Although there is more work to be done, the current manuscript points to important new directions in imaging infectious disease.

Specific comments:

It would be helpful for orientation purposes if the schematics (such as Figure 2a) also depicted the timeline for infection, treatment, and imaging. This would also be helpful in Figure 5. They could define late-stage vs. early-stage treatment up front and the results would be easier to follow. Granted, the information is in the actual figures, but there are 24 h treatment time points and 24 h imaging time points which can be a bit confusing at first.

For completeness, it would have been nice to see LSM or confocal microscopy of the mice that were treated effectively (Fig 5), to confirm inhibition of fungal growth.

Broader questions that could be discussed – are their sub-species or strains of *A.fumigatus* and does this antibody recognize all of them? How well might antibody-based imaging work in nodules that are larger and not vascularized, or deeper lesions in less accessible sites such as the brain?

Reviewer #3 (Remarks to the Author):

The authors have presented a dual-labelled antibody approach to studying invasive pulmonary aspergillosis. My expertise is in fluorescence imaging and so I am not able to comment on the impact or biological/clinical interpretation, relevance or accuracy of this study.

Comments:

I found the text very confusing in terms of the order and timing of the infection, administration of VCZ 24 and tracer injection. Figure 2a shows that treatment (VCZ 24) occurs at the same time as tracer injection, but the text on line 135 suggests that the tracer is injected at the same time as infection. Please clarify.

Following on from the previous comment, line 165 suggests tracer injected at same time as infection.

Line 167 – please state that imaging was done at 48 hrs after infection.

Line 169 “precise” and line 173 “intense”. Both are subjective terms, please replace with non-subjective language.

Line 187 “but nevertheless yielded meaningful analyses.” It is not clear what the meaningful analyses are here as all values are clustered around zero in the corresponding figure 4A.

Line 195 refers to “regardless of VCZ treatment (Fig. 4D)” but information on VCZ treatment is not shown in the figure.

Line 196 – seems arbitrary to choose 15% threshold here. Is the same correlation observed for the 10% threshold?

Line 224 & 229 – it is unclear why the authors change between use of the 10% threshold and the 15% threshold. Please be consistent and state that similar results were obtained with both thresholds.

Line 231 – there is no panel D showing in figure 5.

Lines 465, 466, 474. Please make it clear to the reader that the xxx/yy numbers are centre wavelength in nm and bandpass width in nm.

REVIEWER COMMENTS

We would like to thank the reviewers for their critical feedback, to which we reply below. We believe the changes, corrections and additions they generated have significantly improved the quality of the manuscript and its potential reach to a larger audience.

The reviewer comments will be presented in italics, our comments and answers in regular font.

Reviewer #1 (Remarks to the Author):

The authors have developed an exemplary imaging system for monitoring experimental invasive aspergillosis. They have shown the distribution of infection in the lung after experimental infection, and its progression in a short time frame in neutropenic mice. They contrast direct visualization of the fungus itself with a fluorophore-linked monoclonal antibody, that is used commercially for diagnosis. Over a short time frame (24 hours) they show the impact of voriconazole.

Comments

Methods

1. *L401 The names or numbers of the *A. fumigatus* strains used need to be stated.*

The names of the strains are now stated in the manuscript: the clinical wild-type isolate ATCC 46645 ("wild type *A. fumigatus*") and the same strain with the tdTomato gene inserted via pSK379-Tomato plasmid in the His2 sequence ("*A. fumigatus*^{tdTomato}") (L433-434). The strain was first established by Sven Krappmann and co-workers and was first published here: Lothar, J., et al. (2014). "Human dendritic cell subsets display distinct interactions with the pathogenic mould *Aspergillus fumigatus*." *Int. J. Med. Microbiol* 304(8): 1160-1168.

2. *Do the authors know that both strains of *Aspergillus* that they used were susceptible to voriconazole? There refer to 'wild type', which was clearly pathogenic, but many strains in the environment and from patients in Europe are resistant to azoles. What are the MICs? And have they assessed fungicidal activity for these strains in vitro (the MFCs)? About 60% of *A. fumigatus* strains are killed by voriconazole in vitro (Warn et al, PMID: 16284102). This needs to be done.*

We thank the reviewer for this important suggestion. To address this, we have now performed two susceptibility assays to voriconazole of the used strains to confirm sensitivity to VCZ. These are now referenced in the manuscript in the Materials and Methods (L432-458), Results (L137-146) and Supplementary Materials sections (Supp. Tables 1 and 2, Supp. Fig. 2 and 3). Briefly, MIC determination for voriconazole by visual reading, as suggested in the EUCAST guidelines (Lass-Flörl et al., 2006),

resulted in 0.5 µg/ml for the wt ATCC46645 strain as well as for the *A. fumigatus*^{tdTomato} strain. This equaled the MIC results of the work by Lass-Flörl et al. for different *A. fumigatus* wt strains. To back-up these results with a more sensitive reading method we decided to also analyze the 96-well plates spectrophotometrically following the suggestion of Meletiadis et al. 2017. Results indicated a similar MIC of 0.5 µg/ml for the wt ATCC46645 strain and of 0.25 µg/ml for the *A. fumigatus*^{tdTomato} strain.

General

3. *Abstract and L97 – the phraseology of ‘early intervention’ to ‘prevent’ makes little sense. If infection is established, it cannot be prevented.*

We have replaced “preventing the disease” by “prevent complete invasion of the lungs by the fungus” (L41).

4. *L242. The GM ELISA test was introduced 20 years ago. What is new are the lateral flow assays.*

We have now mentioned the GM-ELISA test appropriately and corrected the sentence (L270-271).

5. *L251 Re other non-invasive tests – consider also Skriba et al (PMID: 30349512) and Marr et al (PMID: 29684106) as new (not yet in the clinic) alternatives, that should be mentioned.*

These omitted valuable articles on non-invasive urine-based tests for IA have now been mentioned in the text (L274).

6. *L279 Consider fungal keratitis detection Lee et al (PMID: 29228372), although not used to follow treatment.*

Thank you for bringing this article to our attention. We have now cited this research in our article, mentioning its advantages but keeping it in context, as the penetration depth of optical imaging remains too low for lung imaging. L296-298: “In parallel, optical detection of *A. fumigatus* has been shown in a fungal keratitis model using fluorescent liposomes, achieving rapid and sensitive diagnosis, although with an imaging depth too limited for lung imaging”.

7. *L304 Not correct. Needs rephrasing. In general, voriconazole ‘holds’ infection in profound neutropenia, until neutrophil recovery occurs. It is not expected that it would eradicate (or even lessen) infection in this short time frame with some help from the immune system. Some isolates are killed by voriconazole, others not.*

We thank the reviewer for this observation. We have referenced in the manuscript several publications referring to the fungicidal effect of Voriconazole *in vitro* on different *A. fumigatus* strains, but have had issues finding relevant data for *in vivo* murine models, in particular within short experimental time frames. In addition to the sensitivity assay addressed above, we have now rewritten this section of the manuscript to better represent the fungistatic activity of the drug (L341-342).

8. *In terms of clinical utility, is the diagnosis of Aspergillus nodules in cases where lung cancer is the main differential diagnosis another possible clinical case? See Baxter et al, PMID: 21460371 and Muldoon et al, PMID: 27538521). The authors mention cases of IA in severe influenza and COVID-*

19, but it not realistic to imagine that such severely ill patients could be taken for PET imaging – another limitation. One of the other challenges clinically is determining when to stop therapy as CT and CXR imaging lags behind clinical cure. Could this technology allow the end point of antifungal therapy to be determined, shortening treatment periods and allowing additional immunosuppressive procedures to be done earlier with greater chances of success?

Thank you for this interesting comment. We took the opportunity to discuss this issue further with our colleagues from the Radiology department of the University Hospital, which hopefully will shed more light on the matter:

Indeed, the differential diagnosis of lung cancer and aspergillosis nodules might be another possible application of our immuno PET, although there are many different reasons for lung nodules and IPA would be a very specific and rare reason for that. Aspergilloma would be another example of lung nodules visible in MRI/CT, with little blood supply (a lump of fungus growing within an air-filled cavity) which would require differential diagnosis from IPA. So, our *Aspergillus* PET procedure would only be recommended if an IPA is suspected. But, in this case, we do not see why differentiation from lung-cancer should be problematic. These are important issues that should be covered in a future large-scale clinical trial.

PET imaging is a complex procedure, but possible also for critically ill patients from intensive care units. Because of the long uptake time, tracer application can be performed at the ward, but local radiation safety regulations as well as radiation exposure for medical staff have to be taken into account. In the particular case of COVID-19 patients and during the review process of our manuscript, new reports have emerged of patients with aspirates yielding a positive LFD with either negative serum galactomannan (Prattes et al. Invasive pulmonary aspergillosis complicating COVID-19 in the ICU - A case report. *Med Mycol Case Rep*, (2020)) or positive serum galactomannan (Meijer et al. Azole-Resistant COVID-19-Associated Pulmonary Aspergillosis in an Immunocompetent Host: A Case Report. *J Fungi (Basel)*, 6, (2020)). In both cases, patients were imaged with X-ray. Thus, JF5 based PET could indeed be an interesting route with high specificity and sensitivity, even in critically ill patients.

Regarding determination of the end point of treatment, we thank the reviewer for mentioning this very important issue. We do not see any principal counter-argument, why JF5-PET should not be able to determine when *Aspergillus* is eradicated and therefore the approach could improve clinical decisions regarding the duration of the antifungal treatment. Of course, interference from antifungal drugs needs to be elucidated in further studies and the diagnostic value has to be validated in prospective clinical trials. This is now better mentioned in the last section of the manuscript (L414-415).

9. Not all cases of IA are caused by A. fumigatus. The species specificity of the JF5 antibody should be stated and if not covering the common infecting species, mentioned as a limitation. In many parts of the world, A. flavus is more common than A. fumigatus and in some institutions A. terreus is particularly common.

JF5 recognizes all clinically relevant *Aspergillus* species. We have now added a sentence to mention this fact, along with a suitable reference (Thornton CR. *Clin Vaccine Immunol* **15**, 1095-1105 (2008)) in the discussion (L314-317).

10. A single sentence comment on brain imaging based on the blood brain barrier penetration of their reagents would be helpful. In irbutinib-treated patients, cerebral aspergillosis much more common proportionately.

David Denning

The reviewer is correct with the assumption that the full JF5 antibody would not be able to cross an intact BBB. However, we would assume some form of BBB breakdown at positions, where the formation of cerebral infectious centers has occurred, e.g. induced by gliotoxin (<https://pubmed.ncbi.nlm.nih.gov/30006720/>) or even by damage from the fungus itself (i.e. cerebral hemorrhage). In these cases, we might also get access of the antibody to the brain nodules and since we always image the entire mouse/patient, it is likely that such antibody enrichments would be detectable in an otherwise signal-negative organ. Matched with highly conspicuous MR or CT features of the brain at such sites, it appears possible that immuno PET would be able to diagnose cases of cerebral IA. This discussion has been added to the MS L317-321).

Reviewer #2 (Remarks to the Author):

Hennenberg et al describe application of in vivo antibody-based imaging to the detection of Aspergillus fumigatus infections, evaluation of extent of disease, and response to treatment. There is no question that Aspergillus infections remain an important clinical challenge, particularly in patients with impaired immunity, lung disease, or as an aftermath of viral illnesses. It is rewarding to see molecular imaging approaches increasingly and effectively being applied in infectious disease. Here, a humanized Aspergillus-specific antibody, JF5, has been dually-labeled (with Cu-64 and DyLight 650) for PET and fluorescence imaging in an aggressive mouse model of disease. The approach is further applied to assess response to treatment with voriconazole (VCZ).

The imaging studies are well executed and supported by corroborative studies. Strengths of the work include validation of the imaging observations via ex vivo microscopy including vivid 3D light-sheet microscopy and confocal microscopy. Furthermore, since the A.fumigatus strain used expresses a red fluorescent protein (Tomato), LSM was used to confirm colocalization of the DyLight-labeled antibody in the mouse lungs. The investigators approach quantitation rigorously, exploring different cut-offs to attempt to establish objective criteria for confirming disease.

- *One of the challenges of the current work is the use of a severe model: mice which have been rendered neutropenic, allowing very rapid proliferation of the fungus. Normal mice resolve disease over 6-24 h, but the neutropenic mice develop persistent, fulminant infection over the 6-48 h time course. Treatment at 24 h is ineffective, but two doses, at 3 + 24 h appears effective. Conclusions drawn would not necessarily be relevant or even feasible in the clinical setting (i.e. starting treatment within 3 h of infection; patients are not infected at a specific time).*

The clinical setting is indeed slightly different from our artificial pre-clinical model as we force the almost complete disappearance of Ly6G⁺ and Ly6C⁺ cells and deposit a high number of spores directly into the lungs to achieve a high level of infection rapidly. The human presentation of the disease usually has a slower progression and allows a larger window of intervention that could ideally be better monitored by our approach. For the imminent clinical translation, we aim at performing imaging of patients at diagnosis / immediately after the first antifungal injection and at least one week later.

- *A significant challenge is quantification of the extent of fungal invasion. The authors take a systematic approach, establishing a threshold of activity in the PET scans that allows them to distinguish healthy vs diseased animals. Radiotracer activity levels of 5%ID/g did not differentiate between healthy and sick; 10% ID/g did identify diseased animals, but a fairly high threshold of 15% ID/g was needed to eliminate a false positive signal in control mice. Using this threshold (15%ID/g), an infection volume could be defined. The necessity of a high threshold can potentially reduce sensitivity. The fundamental problem is the use of an intact antibody with extended persistence in the circulation, leading to high activity in highly vascularized lung tissue (due to blood activity ~20%ID/g). The authors should discuss alternatives such as smaller antibody fragments or small protein scaffolds with more rapid elimination from the circulation. Also, this quantitation scheme is highly specific to the neutropenic mouse model and would need to be re-established for each new model, or for clinical use.*

This is correct. The use of a full-length antibody as a targeting vector extends the blood circulation time of the radiotracer and provides a high background signal in healthy tissues. While it remains possible that, in the case of human IPA, localised nodules rather than diffuse infection of the organ might provide sufficient contrast between diseased and healthy tissue, the antibody fragment approach provides a possible alternative approach to imaging IPA. Indeed, we are currently exploring this as a means to lowering the time of residence of radioactive substances in circulation, while simultaneously increasing the contrast of the images. To this end, we have generated monovalent (scFv, F(ab)) and bivalent (F(ab')₂) fragments of hJF5-based radiotracers, but the data are not mature enough yet to be included in this manuscript. This is now mentioned in the discussion (L483-489) and will be the focus of future studies from our group.

Regarding the quantitation scheme, it indeed needs to be re-established for each new model, which can be done directly from one experiment by including a control and a diseased animal without treatment. For its translation into the clinic, we believe it will be easier than the pre-clinical setting, as the lung will very frequently present a healthy lobe / part of tissue that can be used as an internal reference for thresholding. This, however, will need to be clarified when pursuing the first clinical investigations.

Nonetheless, the ability to visualize disease progression and optimize treatment through non-invasive imaging demonstrates the potential impact of this approach in infectious disease. The use of a pathogen-specific immunoPET probe clearly provides advantages over more general tracers such as 18F-FDG. Although there is more work to be done, the current manuscript points to important new directions in imaging infectious disease.

Specific comments:

- *It would be helpful for orientation purposes if the schematics (such as Figure 2a) also depicted the timeline for infection, treatment, and imaging. This would also be helpful in Figure 5. They could define late-stage vs. early-stage treatment up front and the results would be easier to follow. Granted, the information is in the actual figures, but there are 24 h treatment time points and 24 h imaging time points which can be a bit confusing at first.*

In order to clarify the timelines used for all *in vivo* experiments, we have detailed further the workflow presented in the panels 1a, 2a, 3a, and have added a panel in Fig 5. We hope these amendments now allow better orientation for the reader.

- *For completeness, it would have been nice to see LSM or confocal microscopy of the mice that were treated effectively (Fig 5), to confirm inhibition of fungal growth.*

We did not perform LSM or confocal microscopy of the lungs presented in Fig. 5, since inhibition of fungal growth was, in our view, sufficiently evident from the combined data of mouse weight, PET imaging and *ex vivo* biodistributions. In addition, the organs upon removal could visually be clearly identified as infected or not. However, we performed H&E and Grocott methenamine silver stains of several lung lobes which confirmed the efficiency of early treatment when compared to late treatment,

with early treatment resulting in almost complete clearance of the fungus (few and small *Aspergillus* granulomas were identified, which contained exclusively spores. No hyphae were detected.) and very minimal tissue damage. In contrast, late treatment points showed multiple and large aspergillus granulomas containing numerous and large hyphae, with angioinvasion. Those experiments are now described in Materials and Methods (L567-570), the result section (L255-264) and representative images are shown in Supplementary Materials (Supp. Fig. 5).

- *Broader questions that could be discussed – are their sub-species or strains of A.fumigatus and does this antibody recognize all of them? How well might antibody-based imaging work in nodules that are larger and not vascularized, or deeper lesions in less accessible sites such as the brain?*

We thank the reviewer for mentioning this important point, which has also been made by Reviewer #1. The JF5 antibody recognizes all species of *Aspergillus*, including *A. flavus* and *A. terreus* (as discussed in our response above to Reviewer #1). This is now mentioned in the text with an appropriate citation (L314-317). The access in large clinical nodules that might not be homogeneously vascularized and necrotic at the core, or even in layers, still needs to be assessed, and is currently one of our active research topics. We hope to access human samples after a potential image-guided biopsy of aspergillosis nodules, although we recognize the chances of obtaining such a human sample are quite low. The brain is also a high priority target – however, we do not believe an intact antibody would be able to reach an infection site, as we discussed in a reply to a comment from Reviewer #1, as the disruption of the BBB is likely to be too limited to allow for antibody penetration to the organ. However, as the brain is in the field of view of both our preclinical and clinical imaging protocol, we will be able to see cerebral IA should such an event occur. This is now also briefly mentioned in the discussion (L317-321).

Reviewer #3 (Remarks to the Author):

The authors have presented a dual-labelled antibody approach to studying invasive pulmonary aspergillosis. My expertise is in fluorescence imaging and so I am not able to comment on the impact or biological/clinical interpretation, relevance or accuracy of this study.

Comments:

- I found the text very confusing in terms of the order and timing of the infection, administration of VCZ 24 and tracer injection. Figure 2a shows that treatment (VCZ 24) occurs at the same time as tracer injection, but the text on line 135 suggests that the tracer is injected at the same time as infection. Please clarify.*

We apologize for any confusion that has occurred from our images. To clarify the experimental procedures, we have now generated more explicit workflow figure panels (Fig. 1a, 2a, 3a, 5a), and have amended the text in various places to hopefully remove any remaining uncertainties and potential mistakes. We hope the overall experimental approach and timelines are now clearer to the reviewers and potential readers.

- Following on from the previous comment, line 165 suggests tracer injected at same time as infection.*

During our experiments, the tracer was injected directly (i.e. minutes) after i.t. deposition of the spores (“infection”). This was done in order to have access to later infection time points before animal sacrifice due to weight loss became a necessity.

The sentence has been slightly altered to be clearer: “After infecting neutropenic mice with *A. fumigatus*^{tdTomato} and immediately injecting ⁶⁴Cu-hJF5-DyLight650”, and we believe the addition of more explicit timelines in the figures help alleviate any ambiguity (L177-178).

- Line 167 – please state that imaging was done at 48 hrs after infection.*

“48h after infection” has now been added in the text L179.

- Line 169 “precise” and line 173 “intense”. Both are subjective terms, please replace with non-subjective language.*

We have adjusted the language to more accurately and objectively represent our findings L192, L196.

- Line 187 “but nevertheless yielded meaningful analyses.” It is not clear what the meaningful analyses are here as all values are clustered around zero in the corresponding figure 4A.*

This is a mistake on our part, as this sentence should have been a negative. It has now been corrected: “The highest threshold of 20% ID/cc provided negligible infection volumes of <5% in infected animals, but did not yield significant differences between groups.” L200-201.

- *Line 195 refers to “regardless of VCZ treatment (Fig. 4D)” but information on VCZ treatment is not shown in the figure.*

The data contained in the figure take both from the VCZ-treated mice and the non-treated mice. We have now made the panels 4D and 4E clearer to reflect this, using the same color coding as in the other panels, and have also now explicitly mentioned this in the figure legend.

- *Line 196 – seems arbitrary to choose 15% threshold here. Is the same correlation observed for the 10% threshold?*

The r^2 obtained with a threshold of 10% is much lower, mostly due to the high blood signal, resulting in a background signal close to 10%, yielding false positive regions (now mentioned in L208-209: “which was not the case with a threshold of 10% ($r^2=0.1832$)”). In addition, using a 15% threshold for the %ID/cc PET values provided absolute values comparable to the LSFM measurement, thus a slope for the linear regression of approximately 1 as mentioned in the text. This appeared to us, as the interpretation of *in vivo* PET data that provided the highest correlation with *ex vivo* results, the most likely to represent the pathophysiological situation in the organ. We have now included the r^2 obtained with data points taken from the 10% ID/cc PET threshold results in the text, and have better explained our reasoning for the 15% threshold (L243-247).

- *Line 224 & 229 – it is unclear why the authors change between use of the 10% threshold and the 15% threshold. Please be consistent and state that similar results were obtained with both thresholds.*

The 15% threshold was favored overall due to the data already shown in the previous figure (Fig. 4), where it was found to deliver good correlation between *in vivo* PET data and *ex vivo* LSFM data at the 48h time point. While the 10% threshold showed significant differences between groups only 24h after infection, this was not the case with the 15% threshold (Fig. 5d). We attribute this to the progression of the disease in our model, which is still ongoing at 24h post infection and preventing high accumulation of the tracer in the lungs as well as the circulation time needed to achieve high accumulation of antibodies on their target. At the 48h post-infection time point however, both the 10% and 15% threshold show significant differences between groups. This is now better described in the text.

- *Line 231 – there is no panel D showing in figure 5.*

We apologize for the error in panel nomenclature. Figure 5d is now properly assigned to the 48h volume thresholding of the experiment. The biodistribution panel is now correctly presented and referenced in Supp. Materials. (Supp. Fig. 2).

- *Lines 465, 466, 474. Please make it clear to the reader that the xxx/yy numbers are centre wavelength in nm and bandpass width in nm.*

We have now reformulated the first occurrence of the xxx/yy bandpass filter denomination for clarity (L539-540).

REVIEWER COMMENTS

Reviewer #1 (Remarks to the Author):

The manuscript is much improved.

The MIC data is important - the additional text in the main manuscript results could be reduced in a length a little, if this was desirable from a layout perspective.

Reviewer #2 (Remarks to the Author):

Thank you for the opportunity to review the revised manuscript from Henneberg et al. The revisions address some concerns, but others remain.

In short, this study using an intact antibody labeled with Cu-64, and imaging 48 h later, (and their claim of correlation using LSFM) is a proof of principle, but much work still needs to be done to optimize the radiotracer, to demonstrate clearly what they are imaging, to rigorously address sensitivity and the ability to detect potentially clinically relevant levels of infection.

The authors' inclusion of a more complete schematic of the experimental timeline is helpful but raises additional questions. It was not previously clear at what point the radiotracer was injected, and this is more explicit in the revision.

Concerns:

1) Selection of the combination of intact antibody with Cu-64 (12.7 d half-life). All the PET imaging and quantitation took place at 48 h, which is ~ 4 half-lives after injection. This means only 1/16 of the starting activity remained at the time of imaging. Granted, this still comes to $\sim .78$ MBq/21 microCi, but this would not be realistic for moving forward with clinically. Low counts also mean poor statistics for reconstruction and quantitation.

2) Cumulative aspect of imaging, since radiotracer was injected within a few minutes of fungal inoculation. The development of infection was very rapid – evident at 6 h, treatment started at 3 h and/or 24 h, evaluation at 48 hr. This is not a setting where a slowly-clearing radiotracer would be optimal, because what is seen at the end is the cumulative effect of all these processes. Rather, a smaller, rapid-targeting agent and a short half-life radionuclide would be more informative. What did infection look like at 3 h? at 24 h? at 48 h? None of these questions can be answered with the current radiotracer. It's also not clear whether the 20%ID/g of activity in the blood is still immunoreactive radiotracer, or what. The kinetics over the 48 hours could be very complex. In short, it is not really clear what we're looking at, at 48 hrs.

3) Perfusion. It is not clear whether animals were always perfused or not, and in which studies. Specifically, in the methods - "Ex vivo analysis" (line 509), perfusion with PFA is described, then it is followed by "organs were removed and radioactivity was quantified..." This implies that all the ex vivo biodistributions are from perfused animals. This should be made clear, because it is not standard in the field.

This might also explain discordance between in vivo PET and ex vivo biodistribution at 48 h. Lung uptake in infected and infected/treated animals by in vivo PET appears to be 11-13%ID/g, but by ex vivo counting, even higher – between 30-40%ID/g. (Figure 2 c bottom vs Fig 2 d), while control lung drops. This is finally addressed to a degree in the Discussion (line 370), but it should be explained clearly in the Methods and the legend to Figure 2d, and in the corresponding text in the results, if the mouse was perfused prior to euthanasia and counting for the biodistribution.

It also reinforces the issues with using an intact antibody probe.

5) Blood and normal tissue levels. On page 8, line 167, in the summarizing the results in Figure 2, they state "the circulating blood levelswere similar in all groups..." But in Figure 2d the blood levels in control vs infected are clearly significantly different, as marked by asterisks. This statement should be corrected. Some hypotheses are presented in the Discussion, but this was not followed up experimentally. Elevated blood activity is also noted in the final study (line 249, line 255; Supp Fig 4) but not really discussed. Likewise for the liver in Supp Fig 4, which was $20 \pm 13\%ID/g$ vs $6.9 \pm 2.6\%ID/g$ for late vs early/late treatment. What do they think is the cause for these differences?

This is inaccurately summarized in the Discussion, line 354 "Accumulation of the hJF5 tracer was similar in both treated and non-treated animals, and correlated exclusively with the distribution of the pathogen." The latter half of this sentence is incorrect, as elevated spleen and differences in liver activity were observed among the groups (see above); furthermore, none of this activity was correlated with the distribution of the pathogen.

Have they considered Fc interactions of the antibody? They should conduct parallel studies with an isotype-matched control antibody to examine any antigen-independent uptake driven by Fc:FcR interactions (e.g. on immune cells), which may play a role in increased blood and spleen levels (and might confound interpretation of lung activity).

6) Thresholds. There is still a major concern regarding how the thresholds were set. It would make sense to set a threshold a priori based on some rationale (e.g. lung activity in control animals)(as they suggest in the Discussion, line 396). Instead it reads as if the thresholds were determined after the fact, based on selecting a cut-off that would make their correlations significant. Indeed, in their response, they state “The 15% threshold was favored overall... it was found to delivery good correlation between in vivo PET data and ex vivo LSFM data.” In other words, their underlying assumption is that there was a correlation between the in vivo and ex vivo measurements, and they chose their cut off based on a level that allowed them to show the correlation. This is a weak approach.

Furthermore, whether the threshold is 10% or 15% ID/g to define disease volume, they are throwing away a lot of data and potential sensitivity (esp since control mouse lungs were only 3.9% ID/g in Fig 2d).

They should consult with a statistician familiar with assay development including determination of cut-offs. This includes establishing LOD (limit of detection), LOQ (limit of quantification) and ROC curve analysis. Otherwise, the thresholds are arbitrary and correlations shown are not very useful (and limited this highly artificial neutropenic mouse model).

7) The key experiment is Figure 5, where they show a difference between treating at 3 + 24 h vs 24 h. They show differences by PET using a 10% cut off but there is no confirmation by LSFM.

8) Given the shortcomings of the tracer and the questions regarding quantification and correlation, the potential impact of the current work is of concern.

More broadly, the authors write about the importance of early diagnosis of disease (Abstract, line 34) although they also clearly describe their application as focusing on timing of drug intervention and response to therapy (Abstract line 41; Introduction lines 80 and 89.) Yet they return to the need to “discover lung infection as early as possible...” (line 215) and then promptly return to treatment monitoring in the next sentence. Similarly, the Discussion starts off, “Early and accurate diagnosis of

IPA remains a significant clinical challenge...” (line 267). Here the text continues for a full page before the authors come back to the focus of their current work, therapy monitoring (line 299). They should be consistent about putting their approach in context; the research presented in the manuscript does not address early diagnosis, so much of that discussion, although important, is not relevant to the current work.

In their response they state that in the imminent clinical translation, they plan to image at diagnosis/immediately after the first antifungal injection and at least one week later. How would imaging results at either of these times change the treatment of the patient? Otherwise, potential clinical utility is not clear.

Reviewer #3 (Remarks to the Author):

The authors have addressed all of my comments.

There are just a few minor outstanding points that need to be addressed.

1) Abstract line 39 suggests that light sheet microscopy can be used in vivo. Please clarify that this is only used in situ.

2) Line 53, “exacerbated by...”

3) line 181 “complete overlap” is subjective and suggests that this has been quantified. Please use e.g. “a qualitatively complete”

REVIEWER COMMENTS

We thank the reviewers for their feedback and comments on the manuscript, as they have made the manuscript significantly stronger. We hope our revision will adequately address the recent concerns raised by Reviewer #2.

In the following sections, the reviewer comments will be presented in italics, our comments and answers in regular font.

Reviewer #1 (Remarks to the Author):

The manuscript is much improved.

The MIC data is important - the additional text in the main manuscript results could be reduced in a length a little, if this was desirable from a layout perspective.

Thank you for the feedback. We felt two different approaches to validate MIC data were important, but are ready to shorten the section if required by the editorial team.

Reviewer #2 (Remarks to the Author):

Thank you for the opportunity to review the revised manuscript from Henneberg et al. The revisions address some concerns, but others remain.

In short, this study using an intact antibody labeled with Cu-64, and imaging 48 h later, (and their claim of correlation using LSFM) is a proof of principle, but much work still needs to be done to optimize the radiotracer, to demonstrate clearly what they are imaging, to rigorously address sensitivity and the ability to detect potentially clinically relevant levels of infection.

The authors' inclusion of a more complete schematic of the experimental timeline is helpful but raises additional questions. It was not previously clear at what point the radiotracer was injected, and this is more explicit in the revision.

Thank you for the comments and feedback. We have attempted to address them all comprehensively, and the detailed response can be found below. For some of the remarks pertaining to system accuracy and data analysis, we have solicited the help of Dr. Julia Mannheim, a colleague specialized in quantitation accuracy. Her contribution is now mentioned in the Acknowledgments.

Concerns:

1) Selection of the combination of intact antibody with Cu-64 (12.7 d half-life). All the PET imaging and

quantitation took place at 48 h, which is ~ 4 half-lives after injection. This means only 1/16 of the starting activity remained at the time of imaging. Granted, this still comes to $\sim .78$ MBq/21 microCi, but this would not be realistic for moving forward with clinically. Low counts also mean poor statistics for reconstruction and quantitation.

In our experience (Rolle et al. *PNAS* (2016), Davies et al. *Theranostics* (2017)[relating specifically to mAb JF5], and Beziere et al. (2019) Imaging fibrosis in inflammatory diseases: targeting the exposed extracellular matrix. *Theranostics* 9, 2868-2881), as well as in published data from other groups (such as in Srideshikan et al. (2019) ImmunoPET, [^{64}Cu]Cu-DOTA-Anti-CD33 PET-CT, Imaging of an AML Xenograft Model *Clin Cancer Res* 25, 7463-7474), using 3.7MBq/100 μCi per mouse and quantifying accumulation *in vivo* after 48h has shown this approach to be statistically robust, in particular when the ID/cc reaches over 10%. Our equipment in particular has been thoroughly calibrated and its sensitivity assayed (Mannheim et al. (2012) Quantification accuracy and partial volume effect in dependence of the attenuation correction of a state-of-the-art small animal PET scanner. *Physics in Medicine and Biology* 57, 12) and has shown a linear response to activity changes even at this number of counts. In addition, we perform daily quality control to verify the behavior and stability of the system and its linear response to changes in activity in its field of view, ensuring reproducibility of measurements. Furthermore, the data were reconstructed with a state-of-the-art 3D-OSEM algorithm. Thus, and even if injection at a higher dose is often preferable, the chosen procedure allows for robust statistics for reconstruction and quantitation. The materials and methods section has been updated to reflect these clarifications.

Our experience with clinical compassionate use cases using different ^{64}Cu labeled molecules has shown that 24h and 48h post-injection in the clinical setting is feasible. Also, other groups have claimed optimal time points for quantification of accumulation of ^{64}Cu radiolabeled mAb in humans to be 48h post injection of 130 MBq, albeit in a cancer study using radiolabeled HER2 targeted antibodies (Tamura et al. (2013) ^{64}Cu -DOTA-trastuzumab PET imaging in patients with HER2-positive breast cancer, *J Nucl Med* 54, 1869-1875). Thus, we are confident that given today's scanner sensitivities, detection of an injection of 200+ MBq of ^{64}Cu would allow its detection 48h after injection, in particular in a condition with a % ID/cc above 10. In addition, our Medical Faculty and University just recently decided to buy an ultra-long field of view (ULF) PET/CT system with an increase of sensitivity of a factor 16 compared to classical PET/CT system, further enabling the ^{64}Cu based antibody approach. These ULF systems are now marketed and will soon be installed in many locations world-wide, hence offering this increased sensitivity at many places.

In addition, and as written in the MS L428-431, the human presentation of the disease is nodules of several cm in diameter of dense fungal accumulation against a healthier background, which provides sufficient signal to background accumulation of the radiotracer to be clearly visible. Of course, the exact behavior of the ^{64}Cu -hJF5 tracer in humans remains to be seen, but we currently have no data that would indicate a poorer performance than other similar antibody-based radiotracers already investigated.

2) *Cumulative aspect of imaging, since radiotracer was injected within a few minutes of fungal inoculation. The development of infection was very rapid – evident at 6 h, treatment started at 3 h and/or 24 h, evaluation at 48 hr. This is not a setting where a slowly-clearing radiotracer would be optimal, because what is seen at the end is the cumulative effect of all these processes. Rather, a smaller, rapid-targeting agent and a short half-life radionuclide would be more informative. What did infection look like at 3 h? at 24 h? at 48 h? None of these questions can be answered with the current radiotracer. It's also not clear whether the 20%ID/g of activity in the blood is still immunoreactive radiotracer, or what. The kinetics over the 48 hours could be very complex. In short, it is not really clear what we're looking at, at 48 hrs.*

Point1: long circulating radiotracer:

While it is always desirable to have better tools, we cannot completely dismiss a proven and existing tool because there is still potential for improvement in the future. We are very frequently in discussions with clinical experts about the best way of diagnosing IPA in patients, a problem which still has not been solved satisfactorily. The unified response we receive from these experts is that our immuno-PET approach is the by far best option available at the moment and hence should find its way into the clinic as soon as possible. This is also confirmed by a very recent high-profile review (<https://doi.org/10.1128/CMR.00140-18>.) from J-P Latgé, a highly-respected world expert on aspergillosis, who values our approach as follows: “Novel imaging modalities combining the analytic capability of PET and magnetic resonance imaging (MRI) with *Aspergillus*-specific radiolabeled antibodies, termed immuno-PET/MRI, holds great promise to improve specificity and performance of diagnostics for IPA”.

The use of an intact IgG1 humanised antibody with a long circulation time combined with the short-lived classical murine model of IPA (representing over 80% of all IPA studies) might not be optimal to provide information on the early stages of infection in this particular mouse model, although the information provided at the examined time-points is still representative of the invasion of the fungus as is amply demonstrated when the LSFM and PET results are examined in parallel, and when taking into account the previous studies on the topic published by our group (Rolle et al. *PNAS* (2016) and Davies et al. *Theranostics* (2017)). We agree, as mentioned in the discussion and discussed in the first response to the reviewers, that antibody fragments might provide a quicker tool for diagnosis and therapy monitoring, although this is beyond the scope of this current article. This fragment approach is currently under investigation in our teams, notably the F(ab')₂ and nanobody fragments, but there is absolutely no guarantee that a smaller antibody molecule might improve imaging capabilities. The novel findings described here using the full-length antibody demonstrate the huge potential of our molecular imaging approach for non-invasive *in vivo* diagnosis of IPA and for therapy monitoring of the disease. This justifies further R&D but, as with all biological systems, this remains a long and expensive journey especially under current circumstances. The full IgG also has the advantage of being translatable to the clinical setting (as discussed in Davies et al. (2017)), since human injection of full-length humanised antibodies is a proven and approved methodology. In contrast, injection of antibody fragments or nanobodies would require a full set of studies over many years just to approve the fragment format, hence is not going to be available at short notice.

Point2: circulating radiotracer

Lastly, we believe the remaining 20% ID/g in the blood to be immunoreactive as we have shown that the NODAGA chelation of Cu as well as the antibody are stable in serum (Davies et al. *Theranostics* 2017). We also have tested *in vitro* that the fluorescent labeling by DyLight and ⁶⁴Cu did not interfere with the affinity of the antibody (Supp. Fig. 1), and we have no indication that its stability or immunoreactivity would be affected by circulation *in vivo*. This is now mentioned in the discussion (L382-385).

3) Perfusion. It is not clear whether animals were always perfused or not, and in which studies. Specifically, in the methods - "Ex vivo analysis" (line 509), perfusion with PFA is described, then it is followed by "organs were removed and radioactivity was quantified..." This implies that all the ex vivo biodistributions are from perfused animals. This should be made clear, because it is not standard in the field.

This might also explain discordance between in vivo PET and ex vivo biodistribution at 48 h. Lung uptake in infected and infected/treated animals by in vivo PET appears to be 11-13%ID/g, but by ex vivo counting, even higher – between 30-40%ID/g. (Figure 2 c bottom vs Fig 2 d), while control lung drops. This is finally addressed to a degree in the Discussion (line 370), but it should be explained clearly in the Methods and the legend to Figure 2d, and in the corresponding text in the results, if the mouse was perfused prior to euthanasia and counting for the biodistribution.

It also reinforces the issues with using an intact antibody probe.

Point 1: clarity of the methods and perfusion process.

This is correct – all biodistribution results were obtained from animals after perfusion, performed directly after blood sampling, following the protocol already described in the Materials and Methods section. We have added reminders of this approach in several sections, notably in the results and in figure legends, as needed for ease of comprehension for the reader.

Point2: in vivo versus ex vivo radiotracer accumulation

When comparing the *in vivo* PET quantification results and the biodistribution data, it is indeed clear that there is a significant difference in healthy lungs that we also partially attribute to the extended circulation time of the radiotracer in the blood compartment with lungs being a particularly blood-rich organ. Perfusion washes out the circulating radiotracer, resulting in a signal decrease in biodistribution γ -counting. As for the increase in signal seen in diseased lungs, it is likely that the distribution of the fungal infection in the mouse model leads to a partial volume effect, as mentioned in the discussion. This increase of signal seen *in vitro* compared to *in vivo* results can be seen in our previous articles on the topic (Rolle et al. *PNAS* (2016) and Davies et al. *Theranostics* (2017)) and is seen in the majority of PET experiments (including of other groups). In addition, the density of lungs is low *in vivo*, further making a direct comparison between % ID/cc and % ID/g very difficult.

5) Blood and normal tissue levels. On page 8, line 167, in the summarizing the results in Figure 2, they state “the circulating blood levelswere similar in all groups...” But in Figure 2d the blood levels in control vs infected are clearly significantly different, as marked by asterisks. This statement should be corrected. Some hypotheses are presented in the Discussion, but this was not followed up experimentally. Elevated blood activity is also noted in the final study (line 249, line 255; Supp Fig 4) but not really discussed. Likewise for the liver in Supp Fig 4, which was $20 \pm 13\%ID/g$ vs $6.9 \pm 2.6 \%ID/g$ for late vs early/late treatment. What do they think is the cause for these differences?

This is inaccurately summarized in the Discussion, line 354 “Accumulation of the hJF5 tracer was similar in both treated and non-treated animals, and correlated exclusively with the distribution of the pathogen.” The latter half of this sentence is incorrect, as elevated spleen and differences in liver activity were observed among the groups (see above); furthermore, none of this activity was correlated with the distribution of the pathogen.

Have they considered Fc interactions of the antibody? They should conduct parallel studies with an isotype-matched control antibody to examine any antigen-independent uptake driven by Fc:FcR interactions (e.g. on immune cells), which may play a role in increased blood and spleen levels (and might confound interpretation of lung activity).

Point 1: blood counts

We apologize for the inaccuracy. The differences in blood signal were indeed skimmed over in our manuscript and could have been better explained. We have now put the exact values in the Results section (L171-2). We believe the difference in the values measured between groups is linked to the circulating antigen, an explanation we have now better included in the results (L174) and in the discussion (L382-383).

Point 2: Liver Counts

As for the liver, we do not have a definitive explanation for the difference appearing in certain experiments. We could not see any spores in the organ denoting an invasion of the fungal infection, but we do expect a difference in circulating antigen to yield a difference in radiotracer accumulation in the liver.

Point 3: spleen.

While relatively high spleen uptake is classical in immunoPET studies, in our particular case we believe it is mainly due to specific binding of the antibody to the antigen accumulating in substructures of the spleen (as mentioned L154-155, 372).

Point 4: FcR

We have not looked into Fc/FcR interaction of the antibody extensively as we are using a humanized antibody (based on CDR grafting of a murine antibody on a IgG1 framework) in a neutropenic C57BL/6J mouse. Binding values between human and murine Fc:FcR have been reported elsewhere (Dekkers et al.

2017 Affinity of human IgG subclasses to mouse Fc gamma receptors *Mabs* 9, 767-773) and show that human IgG1 has a K_D of around 1.10^{-7} M to murine Fc γ R, significantly lower than the affinity of the antibody to its target (around 10^{-5} mg/mL as can be extrapolated from the ELISA shown in Supp. Fig. 1). In addition, we have performed binding assays of NODAGA-hJF5 to human peripheral blood mononuclear cells and to human whole blood, using an Alexa Fluor Goat anti-Human antibody and have not found any non-specific binding within the tested concentration range (1-10 μ g/mL).

In addition, the success of the blocking experiments performed with the murine antibody mJF5 were reported in Rolle et al. *PNAS* (2016) and have shown the JF5 affinity for its target antigen *in vivo*; isotype experiments were performed and reported in Davies et al. *Theranostics* (2017) and have shown no significant uptake of a control radiolabeled isotype antibody ET901 (also human IgG1). Thus, we believe the main factor driving the accumulation is the affinity of JF5 for its target, and have therefore not conducted these already published experiments again in our study.

6) Thresholds. There is still a major concern regarding how the thresholds were set. It would make sense to set a threshold a priori based on some rationale (e.g. lung activity in control animals)(as they suggest in the Discussion, line 396). Instead it reads as if the thresholds were determined after the fact, based on selecting a cut-off that would make their correlations significant. Indeed, in their response, they state "The 15% threshold was favored overall... it was found to delivery good correlation between in vivo PET data and ex vivo LSFM data." In other words, their underlying assumption is that there was a correlation between the in vivo and ex vivo measurements, and they chose their cut off based on a level that allowed them to show the correlation. This is a weak approach.

Furthermore, whether the threshold is 10% or 15% ID/g to define disease volume, they are throwing away a lot of data and potential sensitivity (esp since control mouse lungs were only 3.9% ID/g in Fig 2d).

They should consult with a statistician familiar with assay development including determination of cut-offs. This includes establishing LOD (limit of detection), LOQ (limit of quantification) and ROC curve analysis. Otherwise, the thresholds are arbitrary and correlations shown are not very useful (and limited this highly artificial neutropenic mouse model).

Point 1: Threshold rationale and determination.

The main purpose of thresholding was to simply provide an estimation of the volume occupied by the fungus *in vivo* (L194 "In order to quantify fungal load in the lungs *in vivo*"), solving the issue of the high blood signal while providing an insight as to the severity of the infection and potentially increase early diagnostic potential. Our intention was not for the reader to understand that we randomly picked thresholds to obtain a good correlation, but rather that each threshold selection was justified reasonably and found to be worth investigating. Of course, our underlying assumption was indeed that *ex vivo* LSFM and *in vivo* PET volume estimations should provide very comparable values simply because they originate from the same molecule, which weighted on the decision of which threshold to consider for further use.

We have now included these explanations in the Results and in the Discussion section. Briefly, the 5% threshold corresponded roughly to lower limit of the average signal seen in healthy lungs *in vivo* (6.5 ± 1.7 %ID/cc); 10% to right above this average value with its SD; 15% was set after estimation of the average maximum signal obtained in healthy animals (14.3 ± 1.6 % ID/cc); and lastly 20% as a control for extreme values. Further estimation of the infected lung volume by LSFM, using both TdTomato and DyLight signal, pointed at infected volumes similar in values to what was obtained with 15% ID/cc as a threshold for *in vivo* PET signal.

In addition, and still regarding threshold determination, we have followed the recommendations from various scientific papers that have been elegantly summed up in a review article (Foster et al. (2014) A Review on Segmentation of Positron Emission Tomography Images *Comput Biol Med* 50, 76-96.), noting that “there is no general consensus on the selection of a thresholding level (especially automatic threshold selection)”. Our approach falls into their definition of “Fixed thresholding”, in particular the subset “fixed thresholding with user guidance” as we include information from multiple sources to establish a reasonable threshold.

Point 2: throwing away data:

We believe the thresholding only serves to highlight the actual infection foci by removing remaining background signal (from healthy tissue and circulating blood). In all of the animals we have investigated and applied the thresholding method to, we did not see a false positive or a false negative. Furthermore, the thresholding was only applied to *in vivo* PET results (expressed in % ID/cc, so pertaining to a volume), whereas Fig. 2d pertains to *ex vivo* counts (expressed in % ID/g, pertaining to a mass). In terms of clinical translation potential, we believe that this thresholding approach will be manageable directly *in situ* by using different parts of the lungs as the human disease usually presents itself in isolated nodules of several centimeters diameter.

Point 3: LOD, LOQ, ROC

Regarding further statistical analysis, we performed a ROC curve analysis of the volume occupancy at different threshold values but the results only mirrored the results presented in the bar graphs in Fig 4 and 5, showing for the 10% and 15% ID/cc thresholds an area under the curve of 1 (control animal to infected animals), whereas using 5% and 20% ID/cc thresholds provided AUC under 0.9. This approach however only discriminates between healthy and diseased animals and disregards the volume estimation, which is a main objective of the thresholding approach, explaining why we have chosen not to include it in the manuscript. We have however including the ROC curves in Supplementary Materials (Supp. Fig. 4) now to show the robustness of our approach and mentioned its results in the text (L210-13).

7) The key experiment is Figure 5, where they show a difference between treating at 3 + 24 h vs 24 h. They show differences by PET using a 10% cut off but there is no confirmation by LSFM.

We have not performed LSM of these samples but rather have opted for silver stains, the standard procedure for visualisation of fungal elements in lung tissues. This provides a microscopic perspective on the success of treatment by showing no hyphae when treatment started at an early time point. We believe this experiment to be sufficient in proving the main point of figure 5, that is to say the necessity of starting treatment early to achieve therapeutic efficacy. The key point of having the dual label Ab approach was to show what a PET signal actually means in terms of fungal structure labelling. This question is amply answered by Fig. 3. Since the samples are no longer available, redoing these experiments under the current COVID-19-lockdown conditions would be extremely challenging and time consuming while adding little extra information.

8) Given the shortcomings of the tracer and the questions regarding quantification and correlation, the potential impact of the current work is of concern.

More broadly, the authors write about the importance of early diagnosis of disease (Abstract, line 34) although they also clearly describe their application as focusing on timing of drug intervention and response to therapy (Abstract line 41; Introduction lines 80 and 89.) Yet they return to the need to “discover lung infection as early as possible...” (line 215) and then promptly return to treatment monitoring in the next sentence. Similarly, the Discussion starts off, “Early and accurate diagnosis of IPA remains a significant clinical challenge...” (line 267). Here the text continues for a full page before the authors come back to the focus of their current work, therapy monitoring (line 299). They should be consistent about putting their approach in context; the research presented in the manuscript does not address early diagnosis, so much of that discussion, although important, is not relevant to the current work.

In their response they state that in the imminent clinical translation, they plan to image at diagnosis/immediately after the first antifungal injection and at least one week later. How would imaging results at either of these times change the treatment of the patient? Otherwise, potential clinical utility is not clear.

Point 1: early diagnosis versus therapy monitoring

We believe both early diagnosis and therapy monitoring to be equally as important, as symptoms are shared with other diseases and azole-resistant strains are becoming prevalent. Generally speaking, a non-invasive molecular imaging approach providing definitive pathogen identification, such as the one we report here, can be considered “early diagnosis” as classical clinical diagnosis identifying the microorganism takes several days and even weeks in certain hospitals. This delays diagnosis resulting in the empiric use of antifungal drugs, and is contributing to the emergence of azole resistance in clinical strains of the pathogen. As discussed in the first part of the Discussion, a lateral-flow assay which speeds up diagnosis does exist, but relies on bronchoalveolar lavage recovered during invasive bronchoscopy. It also does not provide any information on the extent of the infection. As such, we believe molecular imaging to be a novel early non-invasive diagnostic procedure currently missing from the clinics, as well as a new opportunity toward treatment monitoring. This assumption is also backed up by clinical

colleagues to which we are regularly speaking as well as the most recent reviews on the issue (see response to point 2 of this reviewer).

In terms of prerequisites for imaging techniques, early diagnosis and therapy monitoring are correlated as they heavily rely on the high sensitivity of PET to allow for the detection of small amount of fungus. The work presented in our study has an obvious main application in therapy monitoring as azole treatment is presented, but a significant portion of the work (most notably the thresholding approach for volume quantitation) can be readily applied to early diagnosis, as is shown for example in Fig. 5 (volumes at 24h post infection). We added “early” in L39 to indicate in the abstract the parallel goal of the article, have rewritten parts of the early discussion to better reflect this as well as slightly correcting several sentences through the manuscript to reflect the dual impact on early diagnosis and therapy monitoring of the research reported within (all highlighted). In addition, we have re-arranged/re-written parts of the discussion to avoid artificially drawing the attention of the reader excessively towards early diagnosis.

Point 2: Foreseen clinical application

Lastly, in our previous reply, we commented on the direct application we (in coordination with the clinicians on site) could envision for the current radiolabeled monoclonal antibody on site. Due to patient recruitment, current treatment guidelines for patients with fever of unknown origin and access to the radiotracer, we simply described the most likely scenario. In this instance, we believe that the time points proposed for the scans would provide two benefits: first, the initial PET scan could rule-in/rule-out diagnosis and thus reorient the initial therapeutic choice; second, comparison of a scan at treatment start and after 1 week could provide a very good insight on the efficacy of treatment: a reduction in radiotracer accumulation would indicate a successful therapeutic approach. This has now been made clearer at the end of the discussion.

Reviewer #3 (Remarks to the Author):

The authors have addressed all of my comments.

There are just a few minor outstanding points that need to be addressed.

Thank you again for your constructive feedback.

1) Abstract line 39 suggests that light sheet microscopy can be used in vivo. Please clarify that this is only used in situ.

We have now clarified the abstract by rearranging the sentence: “...can be used in immunoPET/MRI *in vivo* and 3D light sheet fluorescence microscopy *ex vivo* to quantify *A. fumigatus* lung infections and to monitor the efficacy of azole therapy.” and hope to have lifted any confusion.

2) Line 53, *“exacerbated by... ”*

“by” has been added.

3) line 181 *“complete overlap”* is subjective and suggests that this has been quantified. Please use e.g. *“a qualitatively complete”*

This is correct and has been fixed in the text.

REVIEWERS' COMMENTS

Reviewer #2 (Remarks to the Author):

This reviewer appreciates the comprehensive and thoughtful response provided by the authors of this important manuscript. Revisions to the manuscript have addressed all concerns and improve clarity.